# Analytically Computing the Moments of a Conic Combination of Independent Noncentral Chi-Square Random Variables and Its Application for the Extended Cox–Ingersoll–Ross Process with Time-Varying Dimension

Sanae Rujivan [1] , Athinan Sutchada [1] , Kittisak Chumpong [2,3],* and Napat Rujeerapaiboon [4]

[1] Center of Excellence in Data Science for Health Study, Division of Mathematics and Statistics, School of Science, Walailak University, Nakhon Si Thammarat 80161, Thailand
[2] Division of Computational Science, Faculty of Science, Prince of Songkla University, Songkhla 90110, Thailand
[3] Statistics and Applications Research Unit, Faculty of Science, Prince of Songkla University, Songkhla 90110, Thailand
[4] Department of Industrial Systems Engineering and Management, National University of Singapore, Singapore 117576, Singapore
* Correspondence: kittisak.ch@psu.ac.th

**Abstract:** This paper focuses mainly on the problem of computing the $\gamma^{\text{th}}$, $\gamma > 0$, moment of a random variable $Y_n := \sum_{i=1}^n \alpha_i X_i$ in which the $\alpha_i$'s are positive real numbers and the $X_i$'s are independent and distributed according to noncentral chi-square distributions. Finding an analytical approach for solving such a problem has remained a challenge due to the lack of understanding of the probability distribution of $Y_n$, especially when not all $\alpha_i$'s are equal. We analytically solve this problem by showing that the $\gamma^{\text{th}}$ moment of $Y_n$ can be expressed in terms of generalized hypergeometric functions. Additionally, we extend our result to computing the $\gamma^{\text{th}}$ moment of $Y_n$ when $X_i$ is a combination of statistically independent $Z_i^2$ and $G_i$ in which the $Z_i$'s are distributed according to normal or Maxwell–Boltzmann distributions and the $G_i$'s are distributed according to gamma, Erlang, or exponential distributions. Our paper has an immediate application in interest rate modeling, where we can explicitly provide the exact transition probability density function of the extended Cox–Ingersoll–Ross (ECIR) process with time-varying dimension as well as the corresponding $\gamma^{\text{th}}$ conditional moment. Finally, we conduct Monte Carlo simulations to demonstrate the accuracy and efficiency of our explicit formulas through several numerical tests.

**Keywords:** moments; noncentral chi-square random variables; conic combinations; independent; extended Cox–Ingersoll–Ross process; time-varying dimension

**MSC:** 60E05; 60G50; 91G20

## 1. Introduction

Consider a random variable $Y_n$ driven on a probability space $(\Omega, \mathcal{F}, \mathbb{P})$ defined by

$$Y_n := \sum_{i=1}^n \alpha_i X_i \tag{1}$$

for an integer $n \geq 2$, where $\alpha_i > 0$, and each $X_i$ is distributed according to a noncentral chi-square distribution with $\nu_i > 0$ degrees of freedom and a noncentrality parameter $\delta_i \geq 0$, i.e., $X_i \sim \chi^2_{\nu_i}(\delta_i)$, for all $i = 1, \dots, n$. We assume that the $X_i$'s are independent with respect to the $\sigma$-field $\mathcal{F}$ and probability measure $\mathbb{P}$.

This paper focuses mainly on the problem of computing the $\gamma^{\text{th}}$ moment, $\gamma \in \mathbb{R}_+$, of $Y_n$ given by

$$\mathbb{E}[Y_n^\gamma] = \int_0^\infty y^\gamma f_{Y_n}(y)dy \tag{2}$$

where $f_{Y_n}(y)$ is the probability density function (PDF) of $Y_n$, and $\mathbb{E}[X]$ denotes the expected value of a random variable $X$ with respect to the probability measure $\mathbb{P}$. Utilizing the property of the noncentral chi-square random variables [1], we have that $\mathbb{E}[X_i^m]$ is finite for all non-negative integer $m$ and $i = 1, \ldots, n$. This result and the independence of the $X_i$'s ensure the integral on the RHS of (2) is always finite for all $\gamma \in \mathbb{R}_+$.

The random variable $Y_n$ is found in many statistical applications. In hypothesis testing, several test statistics converge in distribution toward a conic combination of independent noncentral chi-square random variables (see, e.g., [2–5]). Moreover, $f_{Y_n}(y)$ and $\mathbb{E}[Y_n^\gamma]$ play an interesting role in financial applications; see, e.g., [6–11]. Very recently, Rujivan and Rakwongwan [11], Chumpong et al. [6], and Rujivan [10] showed that the log-return realized variance when the underlying asset follows the extended Black–Scholes model can be expressed in terms of a conic combination of independent noncentral chi-square random variables. As a result, they derived the exact PDF of the log-return realized variance as well as an explicit formula for the $\gamma^{\text{th}}$ moment of the log-return realized variance for $\gamma = \frac{1}{2}, 1$, yielding the first explicit pricing formulas for volatility swaps, volatility options, variance swaps, and variance options, respectively. Furthermore, Rujivan and Rakwongwan [11] utilized the approach proposed in Rujivan [12] for constructing an approximate formula for pricing volatility swaps for the Heston stochastic volatility model. On the other hand, Rujivan [10] proposed an approximate formula for pricing volatility swaps when the underlying asset evolves according to the constant elasticity of variance model.

We now return to the problem of computing the desired moments. Computing (2) is trivial when the values of the $\alpha_i$'s are equal and $\gamma = m$ is a non-negative integer. In other words, $Y_n$ reduces to a scaled noncentral chi-square random variable, the PDF of which is known, which in turn implies that $\mathbb{E}[Y_n^m]$ can be obtained in an explicit form since calculating the integral on the RHS of (2) can be worked out when $\gamma$ is a non-negative integer (see, e.g., [1,13]). On the other hand, it has been repeatedly shown in literature for several decades, see for example in [13–26], that finding an analytical approach for solving the nonlinear problem (2) is significantly more intricate since the PDF of $Y_n$ is not well-known, when the values of some $\alpha_i$'s are unequal, and it is not clear which existing representations of the PDF lend themselves to the calculation of the moments. This underlines the importance of our study.

Based on the above discussion, our paper has three aims which we now describe. The principal aim is to provide practitioners an accurate and efficient formula for computing $\mathbb{E}[Y_n^\gamma]$ for any integer $n \geq 2$ and $\gamma \in \mathbb{R}_+$, including both integers and nonintegers. The next aim is to illustrate further applications of $Y_n$ in interest rate modeling by adopting a Laguerre expansion for the PDF of $Y_n$ which is proposed in this paper to explicitly derive the transition probability density function (TPDF) of the extended Cox–Ingersoll–Ross (ECIR) process with time-varying dimension and which, to our knowledge, has never been found in explicit form until now. In addition, the ECIR process with time-varying dimension was intensively studied by Maghsoodi [27], where its TPDF was given in explicit form and used for pricing bond options for the case where the dimension is constant. The final aim is to utilize the explicit formula for $\mathbb{E}[Y_n^\gamma]$ obtained in this paper to find a novel formula for computing the $\gamma^{\text{th}}$ conditional moment of the ECIR process which is accurate and more efficient in terms of computational complexity than existing methods, such as those in [12,28].

The rest of the paper is structured as follows. In Section 2, we derive a Laguerre expansion for the PDF of $Y_n$. In Section 3, by utilizing the Laguerre expansion, we write $\mathbb{E}[Y_n^\gamma]$, $n \in \mathbb{N}$, $\gamma \in \mathbb{R}_+$, in terms of generalized hypergeometric functions and analytically estimate its truncation errors. In that section, we also extend our result to computing $\mathbb{E}[Y_n^\gamma]$ when $X_i$ is a conic combination of $Z_i^2$ and $G_i$ in which the $Z_i$'s are distributed

according to normal or Maxwell–Boltzmann distributions while the $G_i$'s are distributed according to gamma, Erlang, or exponential distributions, assuming that the $Z_i$'s and $G_i$'s are independent. Section 4 illustrates a usage of our result in analyzing the ECIR processes. This includes the first explicit formula for the TPDF of the ECIR process with time-varying dimension and novel explicit formula for the $\gamma^{\text{th}}$ conditional moment of the ECIR process. In Section 5, all the explicit formulas proposed in this paper are validated with either Monte Carlo (MC) simulations or other formulas proposed in the literature through several numerical tests. The paper is concluded in Section 6. All proofs are provided in the appendices.

## 2. The PDF of $Y_n$

The PDF of $Y_n$ defined in (1) has been studied by many authors for several decades with various representations (see, for instance, in [16,17,21,24–26]). In this paper, we use the approach proposed in [16] to obtain a Laguerre expansion for the PDF of $Y_n$.

**Theorem 1.** *The PDF of $Y_n$ given in* (1) *can be expressed as*

$$f_{Y_n}(y) = f_{Y_n}^{(\beta)}(y) := \frac{e^{-\frac{y}{2\beta}} y^{\frac{\nu}{2}-1}}{(2\beta)^{\frac{\nu}{2}}} \sum_{k=0}^{\infty} \frac{k!}{\Gamma(\frac{\nu}{2}+k)} c_k L_k^{\left(\frac{\nu}{2}-1\right)}\left(\frac{y}{2\beta}\right) \quad \forall y > 0 \tag{3}$$

*where $\nu := \sum_{i=1}^{n} \nu_i$, $\beta > 0$ can be arbitrarily chosen, $\Gamma(x)$ is the gamma function, and $L_k^{(\eta)}(x)$ is the generalized Laguerre function (see [29]). In addition, $c_k, k = 0, 1, 2, \ldots$, satisfy the recurrent relations:*

$$c_0 = 1 \tag{4}$$

*and*

$$c_k = \frac{1}{k} \sum_{j=0}^{k-1} c_j d_{k-j} \quad \forall k \geq 1, \tag{5}$$

*where*

$$d_1 = -\frac{1}{2\beta} \sum_{i=1}^{n} \delta_i \alpha_i + \frac{1}{2} \sum_{i=1}^{n} \nu_i \left(1 - \frac{\alpha_i}{\beta}\right) \tag{6}$$

*and*

$$d_j = -\frac{j}{2} \left(\frac{1}{\beta}\right)^j \sum_{i=1}^{n} \delta_i \alpha_i (\beta - \alpha_i)^{j-1} + \frac{1}{2} \sum_{i=1}^{n} \nu_i \left(1 - \frac{\alpha_i}{\beta}\right)^j \quad \forall j \geq 2. \tag{7}$$

**Proof.** The proof is provided in Appendix A. $\square$

A couple of remarks should be made about the free parameter $\beta$. First, we note that the impact of $\beta$ goes beyond Equation (3) as it also influences the $c_k$ coefficients through the recurrence relations (4)–(7). Second, the value of $\beta$ can also influence the convergence rate of (3). Indeed, if the $c_k$ coefficients diverge, then it would be more challenging to reliably approximate the infinite sum on the right-hand side of (3) by its truncated version. As a result of this, we follow the procedures of [16] to study and promote appropriate choices of $\beta$ in Section 3.

## 3. The $\gamma^{\text{th}}$ Moment of $Y_n$

Computing $\mathbb{E}[Y_n^\gamma]$ as given in (2) can be achieved with any desired level of accuracy when the PDF of $Y_n$ is explicitly known. In this section, we use the Laguerre expansion (3) to obtain an explicit formula for $\mathbb{E}[Y_n^\gamma]$ as well as to showcase some interesting applications of the Laguerre expansion (3) in the following subsections.

*3.1. Our Explicit Formula for $\mathbb{E}[Y_n^\gamma]$*

From the Laguerre expansion (3) together with some properties of the generalized hypergeometric function

$$
{}_2\mathbf{F}_1(a_1, a_2; b_1; z) = \sum_{k=0}^{\infty} \frac{(a_1)_k (a_2)_k}{(b_1)_k} \frac{z^k}{k!},
$$

where $(\cdot)_k$ denotes the usual Pochhammer symbol, known from [30], and we derive a simple explicit formula for the $\gamma^{\text{th}}$ moment of $Y_n$ for any $\gamma \in \mathbb{R}_+$.

**Theorem 2.** *For any $\gamma \in \mathbb{R}_+$, we have*

$$
\mathbb{E}[Y_n^\gamma] = (2\beta)^\gamma \sum_{k=0}^{\infty} (-1)^k \frac{\Gamma\left(\gamma + k + \frac{\nu}{2}\right)}{\Gamma\left(k + \frac{\nu}{2}\right)} {}_2F_1\left(-k, 1 - k - \frac{\nu}{2}; 1 - k - \frac{\nu}{2} - \gamma; 1\right) c_k \quad (8)
$$

*where the coefficients $c_k, k = 0, 1, \ldots$, are chosen according to (4)–(7), and the parameters $\nu = \sum_{i=1}^{n} \nu_i$ and $\beta > 0$ can be arbitrarily chosen.*

**Proof.** The proof is provided in Appendix B. □

Theorem 2 essentially expresses $\mathbb{E}[Y_n^\gamma]$ in terms of generalized hypergeometric functions. We remark that computing $\mathbb{E}[Y_n^\gamma]$ relies on the $c_k$ coefficients which can be obtained from the recursive Formulas (4)–(7), and we demonstrate later in our numerical study in Section 5 that implementing our Formula (8) for computing $\mathbb{E}[Y_n^\gamma]$ consumes significantly less time and effort than employing MC simulations.

In terms of applications, the result presented in the following corollary can be, for instance, applied to obtain an analytical formula for pricing volatility swaps in the discrete observation case based on the Black–Scholes model with time-varying risk-free interest rate and time-varying volatility as proposed in Theorem 3.1 of Rujivan [10].

**Corollary 1.** *We have*

$$
\mathbb{E}[\sqrt{Y_n}] = \sqrt{2\beta} \frac{\Gamma\left(\frac{\nu+1}{2}\right)}{\Gamma\left(\frac{\nu}{2}\right)} \sum_{k=0}^{\infty} {}_2F_1\left(-k, \frac{\nu+1}{2}; \frac{\nu}{2}; 1\right) c_k \quad (9)
$$

*where the coefficients $c_k, k = 0, 1, \ldots$, are chosen according to (4)–(7) and the parameters $\nu = \sum_{i=1}^{n} \nu_i$ and $\beta > 0$ can be arbitrarily chosen.*

**Proof.** The proof is provided in Appendix B. □

Another interesting special case of Theorem 1 is when $n = 1$, that is, there is only one summand, say $X$. In this case, we can leverage Theorem 1 to compute a noninteger moment of any chi-square random variable as follows.

**Corollary 2.** *For any $X \sim \chi_\nu^2(\delta)$ and $\gamma \in \mathbb{R}_+$, we have*

$$
\mathbb{E}[X^\gamma] = 2^\gamma \sum_{k=0}^{\infty} \frac{\Gamma\left(\gamma + k + \frac{\nu}{2}\right)}{\Gamma\left(k + \frac{\nu}{2}\right)} {}_2F_1\left(-k, 1 - k - \frac{\nu}{2}; 1 - k - \frac{\nu}{2} - \gamma; 1\right) \frac{1}{k!} \left(\frac{\delta}{2}\right)^k. \quad (10)
$$

**Proof.** The proof is provided in Appendix B. □

*3.2. Estimates for Truncation Errors of $\mathbb{E}[Y_n^\gamma]$*

To implement $\mathbb{E}[Y_n^\gamma]$ on a computer, it is necessary to investigate truncation errors, that is, to quantify a loss due to replacing an infinite sum with a finite sum. This subsection

derives an estimate for the truncation errors of $\mathbb{E}[Y_n^\gamma]$ by applying the results proposed in [16] as follows.

To begin with, based on the formula of $\mathbb{E}[Y_n^\gamma]$ in (8), we define

$$E_{k_1,k_2}^{(\gamma)} := (2\beta)^\gamma \sum_{k=k_1+1}^{k_2} (-1)^k \frac{\Gamma\left(\gamma + k + \frac{\nu}{2}\right)}{\Gamma\left(k + \frac{\nu}{2}\right)} {}_2F_1\left(-k, 1 - k - \frac{\nu}{2}; 1 - k - \frac{\nu}{2} - \gamma; 1\right) c_k \quad (11)$$

for any $k_1, k_2 \in \mathbb{N} \cup \{0, \infty\}$ such that $k_1 + 1 < k_2$. Therefore, $E_{K,\infty}^{(\gamma)}$ represents a truncation error of order $K$ of $\mathbb{E}[Y_n^\gamma]$. To estimate this truncation error, we first derive valid bounds for $c_k$ for $k = 1, 2, \ldots$, defined in (5)–(7).

**Lemma 1.** *The coefficient $c_k$, $k \in \mathbb{N}$, satisfies*

$$|c_k| \le e^{\frac{\delta}{2\zeta}} \left(\frac{2k + \nu}{2k}\right)^k \left(\frac{2k + \nu}{\nu}\right)^{\frac{\nu}{2}} \zeta^k, \quad (12)$$

*where $\delta = \sum_{i=1}^n \delta_i$ and $\zeta = \frac{1}{\beta} \max_{i \in \{1,\ldots,n\}} |\beta - \alpha_i|$. Moreover, if $\beta > \frac{1}{2} \max_{i \in \{1,\ldots,n\}} \alpha_i$, then $0 < \zeta < 1$.*

**Proof.** The proof is provided in Appendix B. $\square$

From Lemma 1 above, we further define

$$B_{k_1,k_2}^{(\gamma)}(\zeta) := (2\beta)^\gamma e^{\frac{\delta}{2\zeta}} \sum_{k=k_1+1}^{k_2} b_k(\gamma, \nu, \zeta) \quad (13)$$

for all $k_1, k_2 \in \mathbb{N} \cup \{0, \infty\}$ such that $k_1 + 1 < k_2$ and $\zeta > 0$, where

$$b_k(\gamma, \nu, \zeta) = \frac{\Gamma\left(\gamma + k + \frac{\nu}{2}\right)}{\Gamma\left(k + \frac{\nu}{2}\right)} \left|{}_2F_1\left(-k, 1 - k - \frac{\nu}{2}; 1 - k - \frac{\nu}{2} - \gamma; 1\right)\right| \left(\frac{2k + \nu}{2k}\right)^k \left(\frac{2k + \nu}{\nu}\right)^{\frac{\nu}{2}} \zeta^k. \quad (14)$$

Utilizing Lemma 1, we obtain the following upper bound of a truncation error.

**Theorem 3.** *Supposing that $\beta > \frac{1}{2} \max_{i \in \{1,\ldots,n\}} \alpha_i$, then we have*

$$\left|E_{K,\infty}^{(\gamma)}\right| \le B_{K,\infty}^{(\gamma)}(\zeta) \quad \forall \gamma \in \mathbb{R}_+, k \in \mathbb{N}, \quad (15)$$

*where $\zeta = \frac{1}{\beta} \max_{i \in \{1,\ldots,n\}} |\beta - \alpha_i|$. Furthermore,*

$$\lim_{K \to \infty} E_{K,\infty}^{(\gamma)} = 0. \quad (16)$$

**Proof.** The proof is provided in Appendix B. $\square$

It should be mentioned from Theorem 3 that the inequality $\beta > \frac{1}{2} \max_{i \in \{1,\ldots,n\}} \alpha_i$ ought to hold when we implement the Formula (8) for computing $\mathbb{E}[Y_n^\gamma]$ to ensure that the truncation occurring tends to zero when $K$ approaches infinity.

Finally, we use Euler's transformation [31] in order to show that (8) terminates when $\gamma = m$ is a non-negative integer.

**Theorem 4.** *For any $m \in \mathbb{N}$, we have*

$$\mathbb{E}[Y_n^m] = (2\beta)^m \sum_{k=0}^m (-1)^k \frac{\Gamma\left(m + k + \frac{\nu}{2}\right)}{\Gamma\left(k + \frac{\nu}{2}\right)} {}_2F_1\left(-k, 1 - k - \frac{\nu}{2}; 1 - k - \frac{\nu}{2} - m; 1\right) c_k \quad (17)$$

*where the coefficients $c_k, k = 0, 1, \ldots$, are chosen according to (4)–(7) and the parameters $\nu = \sum_{i=1}^{n} \nu_i$ and $\beta > 0$ can be arbitrarily chosen.*

**Proof.** The proof is provided in Appendix B. □

Applying Corollary 2 and Theorem 4, an explicit formula for the $m^{\text{th}}$ moment of noncentral chi-square random variables can be obtained as follows.

**Corollary 3.** *For any $X \sim \chi_\nu^2(\delta)$ and $m \in \mathbb{N}$, we have*

$$\mathbb{E}[X^m] = m! 2^m \Gamma\left(m + \frac{\nu}{2}\right) \sum_{k=0}^{m} \frac{\left(\frac{\delta}{2}\right)^k}{k!(m-k)!\Gamma\left(k + \frac{\nu}{2}\right)}. \tag{18}$$

**Proof.** The proof is provided in Appendix B. □

*3.3. Analytical Formulas for Other Conic Combinations of Independent Random Variables*

In this subsection, we extend our previous results to various other types of random variables. First, instead of assuming that each component $X_i$ is a chi-square random variable, we assume instead that it is normally distributed with varying means and variances. Second, motivated by its application in queuing theory, we focus on the case where each component follows a gamma distribution with varying shape parameters. Note that gamma distributions are often used in queuing theory for modeling the distribution of certain types of waiting times, e.g., the excess water flow of a dam as explained in Mathai [32] and other problems in communication theory with respect to the performance of certain wireless transmission systems as described in Alouini et al. [33]. Third, we focus on the sum of independent Erlang distributed random variables, which lie at the core of many fields such as telecommunications, statistics, reliability theory, and risk analysis [34]. Last but not least, we focus on the sum of exponential random variables, which are often used in stochastic modeling thanks to its memoryless property, and the sum of squared Maxwell–Boltzmann random variables, which can be used to explain the molecular speed distribution of ideal gases [35], etc.

**Theorem 5.** *Consider a random variable $Y_{(1,n)} := \sum_{i=1}^{n} a_{(1,i)} Z_i^2$, where each $a_{(1,i)} > 0$ and each $Z_i$ is a normal random variable with mean $\mu_{(1,i)} \in \mathbb{R}$ and variance $\sigma_{(1,i)}^2$ for $\sigma_{(1,i)} > 0$. Assuming that all summands are independent, then the PDF of $Y_{(1,n)}$, $\mathbb{E}[Y_{(1,n)}^\gamma]$, and $\mathbb{E}[Y_{(1,n)}^m]$ can be computed using (3), (8), and (17), respectively, for all $\gamma \in \mathbb{R}_+$ and integer $m \in \mathbb{N}$, by setting $\alpha_i = a_{(1,i)} \sigma_{(1,i)}^2$, $\nu_i = 1$, and $\delta_i = \left(\frac{\mu_{(1,i)}}{\sigma_{(1,i)}}\right)^2$ for all $i = 1, \ldots, n$.*

**Proof.** The proof is provided in Appendix B. □

**Theorem 6.** *Consider a random variable $Y_{(2,n)} := \sum_{i=1}^{n} a_{(2,i)} G_i$, where each $a_{(2,i)} > 0$ and each $G_i$ is distributed according to a gamma distribution with shape parameter $\kappa_{(2,i)} > 0$ and scale parameter $\theta_{(2,i)} > 0$. Assuming all summands are independent, then the PDF of $Y_{(2,n)}$, $\mathbb{E}[Y_{(2,n)}^\gamma]$, and $\mathbb{E}[Y_{(2,n)}^m]$ can be computed using (3), (8), and (17), respectively, for all $\gamma \in \mathbb{R}_+$ and integer $m \in \mathbb{N}$, by setting $\alpha_i = \frac{1}{2} a_{(2,i)} \theta_{(2,i)}$, $\nu_i = 2\kappa_{(2,i)}$, and $\delta_i = 0$ for all $i = 1, \ldots, n$.*

**Proof.** The proof is provided in Appendix B. □

**Theorem 7.** *Consider a random variable $Y_{(3,n)} := \sum_{i=1}^{n} a_{(3,i)} L_i$, where each $a_{(3,i)} > 0$ and each $L_i$ is distributed according to an Erlang distribution with shape parameter $\kappa_{(3,i)} \in \{1, 2, \ldots\}$ and rate parameter $\lambda_{(3,i)} > 0$. Assuming that all summands $L_i$ are independent, then the PDF of $Y_{(3,n)}$,*

$\mathbb{E}[Y_{(3,n)}^{\gamma}]$, and $\mathbb{E}[Y_{(3,n)}^{m}]$ can be computed using (3), (8), and (17), respectively, for all $\gamma \in \mathbb{R}_+$ and integer $m \in \mathbb{N}$, by setting $\alpha_i = \frac{a_{(3,i)}}{2\lambda_{(3,i)}}$, $\nu_i = 2\kappa_{(3,i)}$, and $\delta_i = 0$ for all $i = 1, \dots, n$.

**Proof.** The proof is provided in Appendix B. $\square$

**Theorem 8.** *Consider a random variable* $Y_{(4,n)} := \sum_{i=1}^{n} a_{(4,i)} P_i$, *where each* $a_{(4,i)} > 0$ *and each* $P_i$ *is distributed according to an exponential distribution with rate parameter* $\lambda_{(4,i)} > 0$. *Assuming that all summands are independent, then the PDF of* $Y_{(4,n)}$, $\mathbb{E}[Y_{(4,n)}^{\gamma}]$, *and* $\mathbb{E}[Y_{(4,n)}^{m}]$ *can be computed using* (3), (8), *and* (17), *respectively, for all* $\gamma \in \mathbb{R}_+$ *and integer* $m \in \mathbb{N}$, *by setting* $\alpha_i = \frac{a_{(4,i)}}{2\lambda_{(4,i)}}$, $\nu_i = 2$, *and* $\delta_i = 0$ *for all* $i = 1, \dots, n$.

**Proof.** The proof is provided in Appendix B. $\square$

**Theorem 9.** *Consider a random variable* $Y_{(5,n)} := \sum_{i=1}^{n} a_{(5,i)} W_i^2$, *where each* $a_{(5,i)} > 0$ *and each* $W_i$ *is distributed according to a Maxwell–Boltzmann distribution with parameter* $\phi_{(5,i)} > 0$. *Assuming that all* $W_i$'s *are independent, then the PDF of* $Y_{(5,n)}$, $\mathbb{E}[Y_{(5,n)}^{\gamma}]$, *and* $\mathbb{E}[Y_{(5,n)}^{m}]$ *can be computed using* (3), (8), *and* (17), *respectively, for all* $\gamma \in \mathbb{R}_+$ *and integer* $m \in \mathbb{N}$, *by setting* $\alpha_i = a_{(5,i)}\phi_{(5,i)}^2$, $\nu_i = 3$, *and* $\delta_i = 0$ *for all* $i = 1, \dots, n$.

**Proof.** The proof is provided in Appendix B. $\square$

## 4. Extensions to the ECIR Process with Time-Varying Dimension

The ECIR process is one of the most widely used processes to model interest rates and to price financial products such as zero-coupon bond, ex-coupon, moment swaps, options, and interest rate swaps. With time-dependent parameters, the ECIR process is capable of accounting for side information from potential political or economic events. Formally, according to Maghsoodi [27], the ECIR process, denoted by $V_t$, satisfies

$$dV_t = \kappa(t)(\theta(t) - V_t)dt + \sigma(t)\sqrt{V_t}dW_t \tag{19}$$

for $t \in (0, T]$ and $T > 0$ with an initial value $V_0 = v_0 > 0$, where the parameter functions $\theta(t) > 0$, $\kappa(t) > 0$, and $\sigma(t) > 0$ are continuous on $[0, T]$, and $W_t$ is a standard Brownian motion under a probability space $(\Omega, \mathcal{F}, \mathbb{P})$ with a filtration $(\mathcal{F}_t)_{0 \leq t \leq T}$. Note that $V_t$ reduces to a plain-vanilla Cox–Ingersoll–Ross process (CIR process) [36] provided that the relevant parameter functions are constants.

Focusing on the ECIR process (19), we define the dimension of $V_t$ as

$$\mathbf{d}(t) := \frac{4\kappa(t)\theta(t)}{\sigma^2(t)} \tag{20}$$

for $t \in [0, T]$, and this quantity plays an important role for deriving an expression for the distribution of $V_t$. Maghsoodi [27] discovered that when $\mathbf{d}(t) = d \geq 2$ for all $t \in [0, T]$ that included the CIR process, $V_t$ never hit zero almost surely and was in fact a scaled time-changed squared Bessel process; as a result, the TPDF of $V_t$ was explicitly given.

Moreover, Maghsoodi [27] showed that $V_t$ could be represented as a lognormal process through a stochastic time-change when $\mathbf{d}(t) \geq 2$ for all $t \in [0, T]$, but the TPDF of $V_t$ was not analytically derived. Consequently, it has been an open question until now how the TPDF of $V_t$ can be obtained in explicit form when $\mathbf{d}(t)$ is time-varying, based on the stochastic time-varying lognormal process representation.

To demonstrate our contribution in the current paper for solving this problem, we apply our previous results in Sections 2 and 3 to explicitly derive the TPDF of $V_t$ as well as its $\gamma^{\text{th}}$ conditional moment when $\mathbf{d}(t)$ is time-varying, provided that the following two assumptions hold:

**Assumption 1.** $\mathbf{d}(t) \geq 2$ *for all* $t \in [0, T]$.

**Assumption 2.** *The derivative* $\mathbf{d}^{(1)}(t)$ *of* $\mathbf{d}(t)$ *with respect to* $t$ *satisfies* $0 \leq \mathbf{d}^{(1)}(t) < \infty$ *for all* $t \in [0, T]$.

*4.1. The Exact TPDF of the ECIR Process with Time-Varying Dimension*

To realize our objective, we firstly define a parameter function as follows:

$$\tau(t, s) := \frac{1}{4} \int_s^t \sigma^2(\zeta) e^{-\int_\zeta^t \kappa(u)du} d\zeta \tag{21}$$

for $0 \leq s \leq t \leq T$.

Peng and Schellhorn [37] showed that the ECIR process $V_t$ described by (19) could be represented as that of a convergent series of weighted independent noncentral chi-square and chi-square random variables. For the sake of completeness, we summarize their results in the following theorem.

**Theorem 10.** *Supposing that Assumptions 1 and 2 hold, then the ECIR process $V_t$ described by (19) with an initial value $v_0 > 0$ can be expressed as*

$$V_t \overset{law}{=} \lim_{n \to \infty} \sum_{i=1}^n \hat{\alpha}_i \hat{X}_i \tag{22}$$

*for any* $t \in (0, T]$, *where the random variables $\hat{X}_i$ are independent and distributed according to noncentral chi-square and chi-square distributions as*

$$\hat{X}_i \sim \chi^2_{\hat{v}_i}(\hat{\delta}_i), \tag{23}$$

*with the coefficients and parameters in (22) and (23) given by*

$$\hat{\alpha}_i = \tau\left(t, (i-1)\frac{t}{n}\right) \quad \forall i \in \{1, \ldots, n\}, \tag{24}$$

$$\hat{v}_1 = \mathbf{d}(0), \tag{25}$$

$$\hat{\delta}_1 = \frac{v_0}{\tau(t, 0)} e^{-\int_0^t \kappa(u)du}, \tag{26}$$

*and*

$$\hat{v}_i = \mathbf{d}^{(1)}\left((i-1)\frac{t}{n}\right)\frac{t}{n} \quad \forall i \in \{2, \ldots, n\}, \tag{27}$$

$$\hat{\delta}_i = 0 \quad \forall i \in \{2, \ldots, n\}. \tag{28}$$

*In particular, if* $\mathbf{d}(s) = d \geq 2$ *for all* $s \in [0, t]$, *then*

$$V_t \sim \tau(t, 0) \cdot \chi^2_d\left(\frac{v_0}{\tau(t, 0)} e^{-\int_0^t \kappa(u)du}\right) \tag{29}$$

**Proof.** See Theorem 3.1 in Peng and Schellhorn [37]. □

Peng and Schellhorn [37] also represented the TPDF of $V_t$ in terms of a limit of a sequence of convolutions of the PDFs of scaled noncentral chi-square and chi-square random variables.

Instead of utilizing the convolution property for independent random variables as shown by Peng and Schellhorn [37], we apply Theorem 1 to obtain the first explicit formula for the TPDF of $V_t$ with time-varying dimension $\mathbf{d}(t)$.

**Theorem 11.** *The TPDF of $V_t$ defined by*

$$f_{V_t}(v, t|v_0) := \mathbb{P}(V_t = v | V_0 = v_0) \tag{30}$$

*for $v, v_0 > 0$ and $t \in (0, T]$ can be expressed as*

$$f_{V_t}(v, t|v_0) = \frac{e^{-\frac{v}{2\tau(t,0)}} v^{\frac{d(t)}{2}-1}}{(2\tau(t,0))^{\frac{d(t)}{2}}} \sum_{k=0}^{\infty} \frac{k!}{\Gamma(\frac{d(t)}{2}+k)} \hat{c}_k(t, v_0) L_k^{\left(\frac{d(t)}{2}-1\right)} \left(\frac{v}{2\tau(t,0)}\right) \tag{31}$$

*where*

$$\hat{c}_0(t, v_0) = 1, \tag{32}$$

$$\hat{c}_k(t, v_0) = \frac{1}{k} \sum_{j=0}^{k-1} \hat{c}_j(t, v_0) \hat{d}_{k-j}(t, v_0) \quad \forall k \in \mathbb{N}, \tag{33}$$

*and*

$$\hat{d}_1(t, v_0) = -\frac{1}{2\tau(t,0)} v_0 e^{-\int_0^t \kappa(u)du} + \frac{1}{2} \int_0^t \mathbf{d}^{(1)}(s) \left(1 - \frac{\tau(t,s)}{\tau(t,0)}\right) ds, \tag{34}$$

$$\hat{d}_j(t, v_0) = \frac{1}{2} \int_0^t \mathbf{d}^{(1)}(s) \left(1 - \frac{\tau(t,s)}{\tau(t,0)}\right)^j ds \quad \forall j \in \mathbb{N} \setminus \{1\}. \tag{35}$$

*In particular, if $\mathbf{d}(s) = d \geq 2$ for all $s \in [0, t]$, then*

$$\hat{c}_k(t, v_0) = \left(-\frac{e^{-\int_0^t \kappa(u)du}}{2\tau(t,0)}\right)^k \frac{v_0^k}{k!} \quad \forall k \in \mathbb{N} \cup \{0\}. \tag{36}$$

**Proof.** The proof is provided in Appendix C. □

*4.2. The $\gamma^{th}$ Conditional Moment of the ECIR Process with Time-Varying Dimension*

For $\gamma \in \mathbb{R}_+$ and a probability space $(\Omega, \mathcal{F}, \mathbb{P})$ with a filtration $(\mathcal{F}_t)_{0 \leq t \leq T}$, we define the $\gamma^{\text{th}}$ conditional moment of the ECIR process $V_t$ as

$$U_E^{(\gamma)}(t|v_0) := \mathbb{E}^{\mathbb{P}}\left[V_t^{\gamma}|\mathcal{F}_0\right] = \mathbb{E}^{\mathbb{P}}\left[V_t^{\gamma}|V_0 = v_0\right] = \int_0^{\infty} v^{\gamma} f_{V_t}(v, t|v_0) dv \tag{37}$$

for $t \in (0, T]$ and $v_0 > 0$, where $f_{V_t}(v, t|v_0)$ is the TPDF of $V_t$ given in (31).

Rujivan [12] first presented a recursive formula for computing $U_E^{(\gamma)}(t|v_0)$ using a partial differential equation (PDE) approach. Alternatively, we apply Theorems 2, 4, and 11 to obtain a novel explicit formula for $U_E^{(\gamma)}(t|v_0)$ in the following theorem.

**Theorem 12.** *Supposing that Assumptions 1 and 2 hold, then for any $\gamma \in \mathbb{R}_+$ we have*

$$U_E^{(\gamma)}(t|v_0)$$

$$= (2\tau(t,0))^{\gamma} \sum_{k=0}^{\infty} (-1)^k \frac{\Gamma\left(\gamma + k + \frac{d(t)}{2}\right)}{\Gamma\left(k + \frac{d(t)}{2}\right)} {}_2F_1\left(-k, 1-k-\frac{d(t)}{2}; 1-k-\frac{d(t)}{2}-\gamma; 1\right) \hat{c}_k(t, v_0) \tag{38}$$

*for $t \in (0, T]$ and $v_0 > 0$, where $\hat{c}_k(t, v_0), k = 0, \ldots,$ are given in (32) and (33).
In particular, for any integer $m \in \mathbb{N}$,*

$$U_E^{(m)}(t|v_0)$$

$$= (2\tau(t,0))^m \sum_{k=0}^{m} (-1)^k \frac{\Gamma\left(m + k + \frac{d(t)}{2}\right)}{\Gamma\left(k + \frac{d(t)}{2}\right)} {}_2F_1\left(-k, 1-k-\frac{d(t)}{2}; 1-k-\frac{d(t)}{2}-m; 1\right) \hat{c}_k(t, v_0) \tag{39}$$

*for $t \in (0, T]$ and $v_0 > 0$.*

**Proof.** The proof is provided in Appendix C. □

4.2.1. Comparison with Other Formulas

To date, the computation of the conditional moment has only been partially solved due to the unavailability of the transitional PDF. Indeed, the problem of computing the integral on the RHS of (37) with any stochastic differential equation (SDE) is typically addressed by the Feynman–Kac theorem, where the partial differential equation (PDE) is solved analytically, and some combinatorial techniques are used to simplify the system of recursive ordinary differential equations (ODEs) associated with the conditional moment; see, for instance, [38–41], for more details.

For a more concrete comparison, Rujivan [12] presented the first explicit formula for the $\gamma^{\text{th}}$ conditional moment of the ECIR process (19) with time-varying dimension as a power series. Rujivan demonstrated the effectiveness of this analytical approach over the other state-of-the-art techniques including the method by Dufresne [28] and MC simulations. This result has some similarities and dissimilarities to our work, which we shall explain. Our formula (38) expresses the $\gamma^{\text{th}}$ conditional moment of the ECIR process as an infinite series where the coefficients $\hat{c}^k(t, v_0)$ can be computed analytically recursively. This offers a more efficient way than, say, using Equation (2.2) of Rujivan [12], which can achieve the same purpose of characterizing the conditional moment of the ECIR process. Therein, the parameters must also be computed recursively but the result of each iteration does not have a closed form. The method of Rujivan [12] is therefore more time-consuming and more prone to numerically accumulating errors.

**5. Numerical Results and Discussions**

As shown in Sections 2–4, our theoretical frameworks presented in this paper produce several new explicit formulas for computing the PDF of $Y_n$ defined in (1) and its moments, including the TPDF of the ECIR process (19) with time-varying dimension and its conditional moments. A natural question that may be raised by practitioners is whether these newly derived explicit formulas are accurate and efficient, especially considering that an infinite sum has to be truncated. Therefore, we intensively investigated the accuracies of our explicit formulas to confirm that there were no algebraic errors in the derivation processes as well as to demonstrate the efficiencies of our explicit formulas compared with either MC simulations or other explicit formulas proposed in the literature through a series of numerical examples which were coded in MATHEMATICA 11 and executed on a notebook with the following specifications: Intel(R) Core (TM) i5-6500, CPU @3.20GHz, 16GB RAM, Windows 10, 64 bit operating system.

*5.1. The accuracy of Our Explicit Formula for $f_{Y_n}^{(\beta)}(y)$*

In order to illustrate the accuracy of our explicit Formula (3), we introduced random variables as follows. For any $n \in \mathbb{N}$, we defined

$$Y_n^{(j)} := \sum_{i=1}^{n} \alpha_i^{(j)} X_i^{(j)} \tag{40}$$

for $j = 1, 2, 3$, where

$$X_i^{(j)} \sim \chi_{v_i^{(j)}}^2 \left( \delta_i^{(j)} \right) \tag{41}$$

for $i = 1, \ldots, n$, and the parameters were set as

$$\alpha_i^{(1)} = \alpha_i^{(2)} = \alpha_i^{(3)} = \frac{2i\alpha}{n(n+1)}, \tag{42}$$

$$v_i^{(1)} = v_i^{(2)} = v_i^{(3)} = \frac{i+3}{2}, \tag{43}$$

$$\delta_i^{(1)} = \frac{i}{10}, \delta_i^{(2)} = 0, \delta_i^{(3)} = \frac{(1-(-1)^i)i}{20}, \tag{44}$$

for $\alpha > 0$. Furthermore, for $n \geq 2$, we assumed that $X_i^{(j)}, i = 1, \ldots, n$ were independent for all $j = 1, 2, 3$. By construction, we note that $\alpha$ represents the total conic coefficients, i.e., $\alpha = \sum_{i=1}^n \alpha_i^{(j)}$, and that each $Y_n^{(j)}$ constitutes a sum of independent noncentral chi-square random variables. Though, we note that through the transformations studied in Theorems 5–9, the distribution of $Y_n^{(j)}$ may be identical to those of other random sums. For instance,

$$Y_n^{(2)} \overset{law}{=} \sum_{i=1}^n \frac{2\alpha_i^{(2)}}{\theta_{(2,i)}} G_i \overset{law}{=} \sum_{i=1}^n 2\alpha_i^{(2)} \lambda_{(3,i)} L_i,$$

where the $G_i$'s are independent random variables and are distributed according to a gamma distribution with the shape parameter $v_i^{(2)}/2$ and the scale parameter $\theta_{(2,i)} > 0$, and similarly the $L_i$'s are independent random variables and are distributed according to an Erlang distribution with the shape parameter $v_i^{(2)}/2$ and the rate parameter $\lambda_{(3,i)} > 0$.

**Example 1.** *We started by considering the PDF of $Y_6^{(1)} = \sum_{i=1}^6 \alpha_i^{(1)} X_i^{(1)}$ in which the values of parameters $\alpha_i^{(1)}, v_i^{(1)},$ and $\delta_i^{(1)}$ for $i = 1, \ldots, 6$ are plotted in Figure 1a. The PDFs of $\alpha_i^{(1)} X_i^{(1)}, i = 1, \ldots, 6$ which varied in range and shape, are shown in Figure 2a–f, respectively. The problem of analytically computing the PDF of $Y_6^{(1)}$ was investigated as follows.*

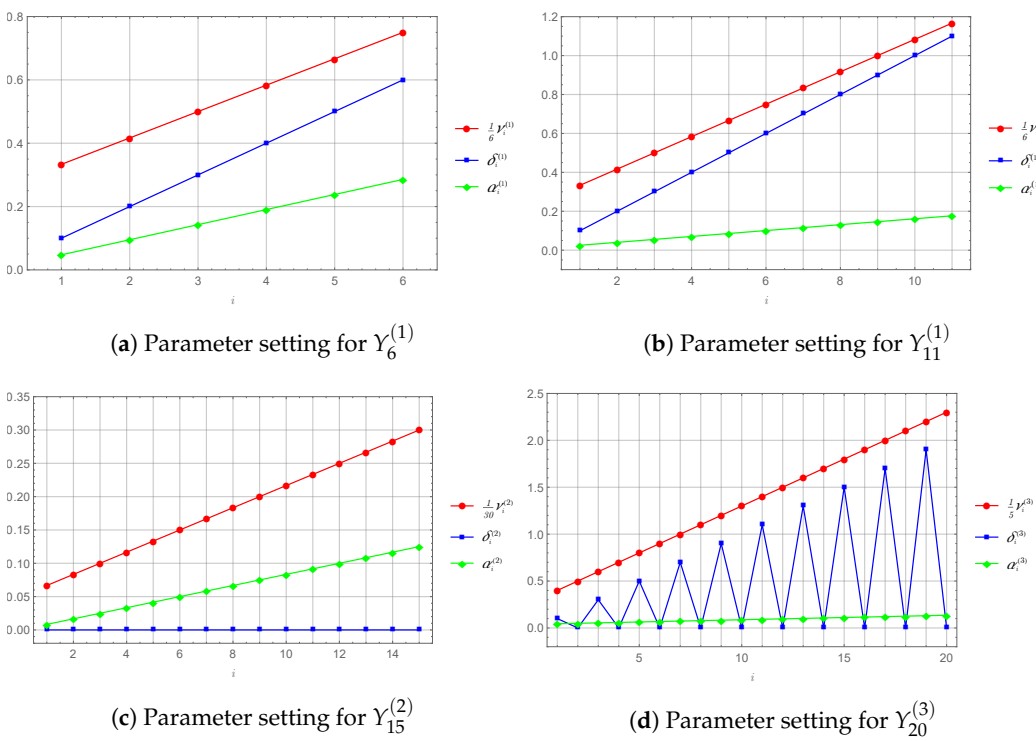

(**a**) Parameter setting for $Y_6^{(1)}$

(**b**) Parameter setting for $Y_{11}^{(1)}$

(**c**) Parameter setting for $Y_{15}^{(2)}$

(**d**) Parameter setting for $Y_{20}^{(3)}$

**Figure 1.** The values of the parameters set in Example 1, in which we plotted $c_j v_i^{(j)}$ with a scaling factor $0 < c_j < 1$ to make its scale comparable to the other parameters for all $j = 1, 2, 3$.

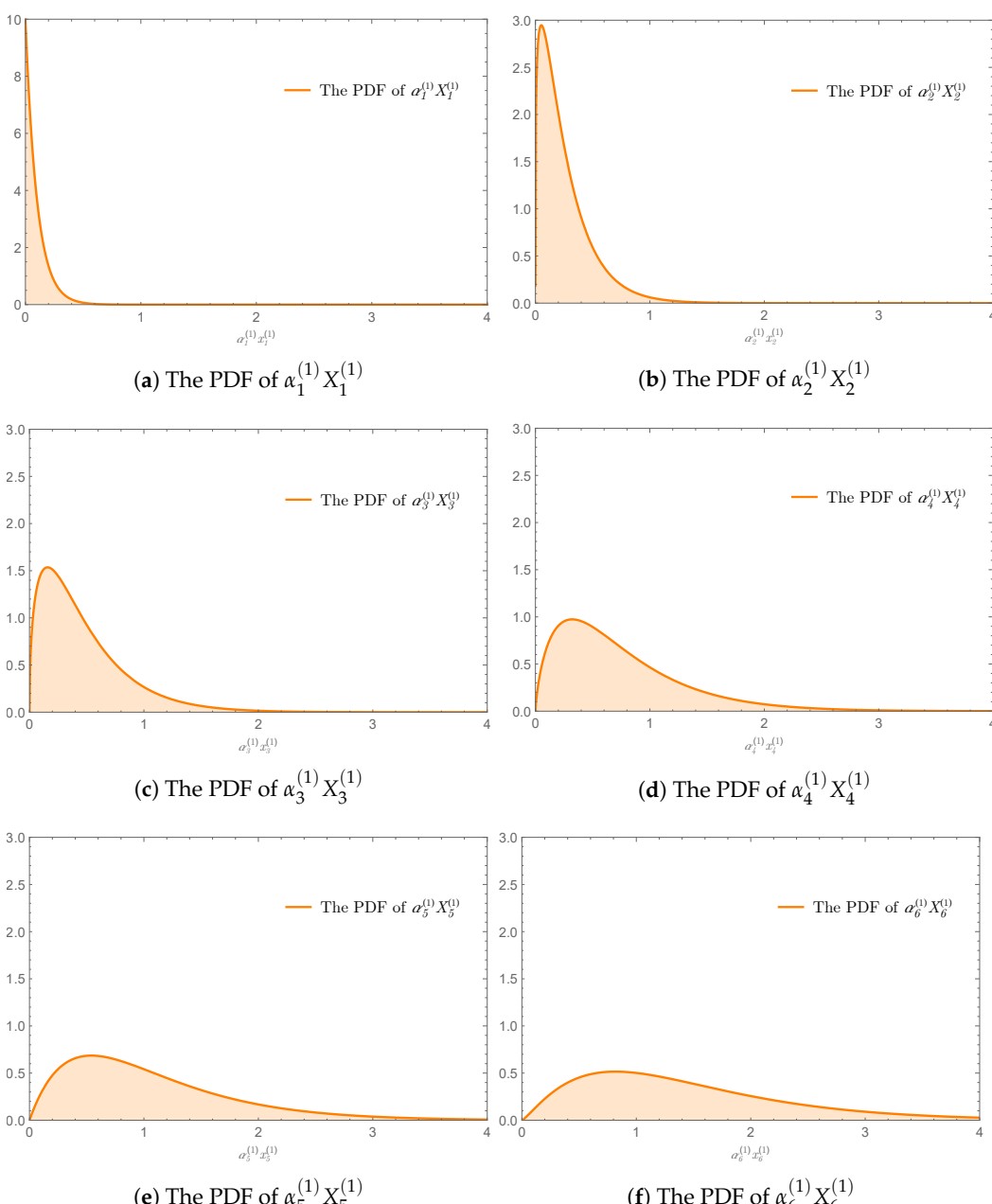

**(a)** The PDF of $\alpha_1^{(1)} X_1^{(1)}$

**(b)** The PDF of $\alpha_2^{(1)} X_2^{(1)}$

**(c)** The PDF of $\alpha_3^{(1)} X_3^{(1)}$

**(d)** The PDF of $\alpha_4^{(1)} X_4^{(1)}$

**(e)** The PDF of $\alpha_5^{(1)} X_5^{(1)}$

**(f)** The PDF of $\alpha_6^{(1)} X_6^{(1)}$

**Figure 2.** The PDFs of $\alpha_i^{(1)} X_i^{(1)}$ in which $X_i^{(1)} \sim \chi^2_{\nu_i^{(1)}}\left(\delta_i^{(1)}\right)$ for $i = 1, \ldots, 6$, and the values of parameters $\alpha_i^{(1)}, \nu_i^{(1)}$, and $\delta_i^{(1)}$ are displayed in Figure 1a.

To obtain the PDF of $Y_6^{(1)}$, denoted by $f_{Y_6^{(1)}}^{(\beta)}(y)$, we set $n = 6, \alpha_i = \alpha_i^{(1)}, \nu_i = \nu_i^{(1)}$, and $\delta_i = \delta_i^{(1)}$ for $i = 1, \ldots, 6$ in (3). The $c_k$ coefficients of the Laguerre expansion (3) were computed by using (4)–(7) with $\beta > \frac{1}{2} \max_i \alpha_i$. Then, the sequence of $c_k$'s was plotted (Figure 3a) showing that $c_k \to 0$ as $k \to \infty$. After inserting the values of the $c_k$'s into (3), a graph of $f_{Y_6^{(1)}}^{(\beta)}(y)$ was displayed, against the histogram of $Y_6^{(1)}$ obtained from MC simulations, as shown in Figure 4a. We clearly see from the figure that the histogram representing the PDF of $Y_6^{(1)}$ perfectly fit the graph of $f_{Y_6^{(1)}}^{(\beta)}(y)$ computed by using the Laguerre expansion (3).

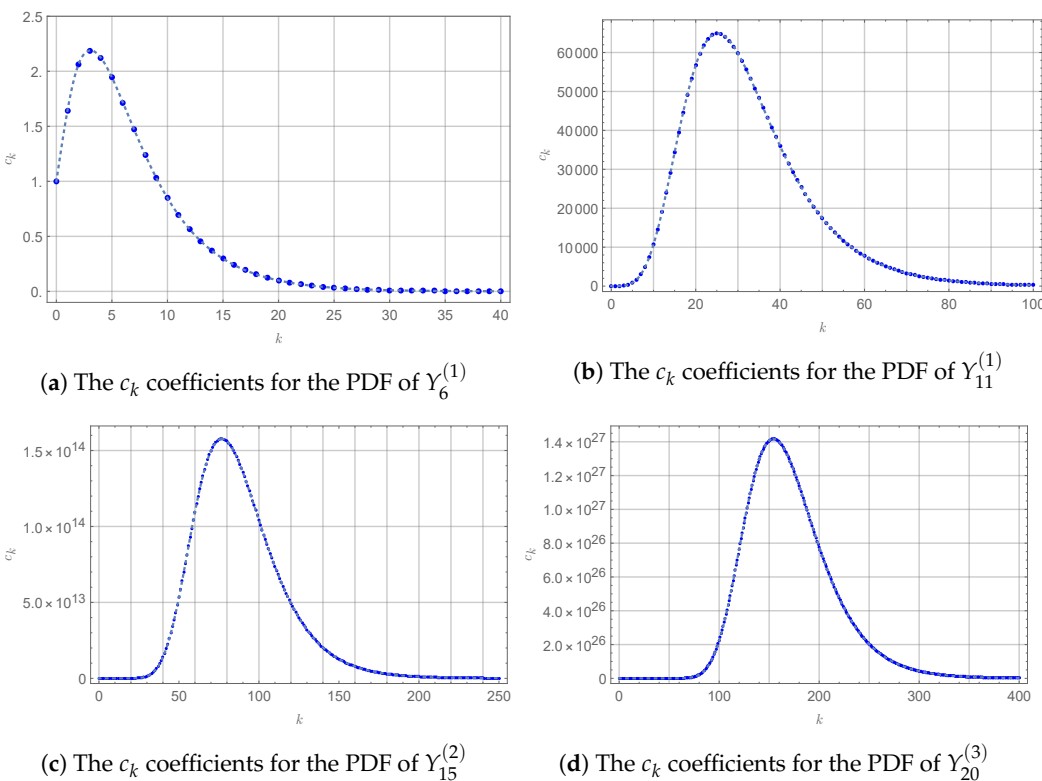

**(a)** The $c_k$ coefficients for the PDF of $Y_6^{(1)}$

**(b)** The $c_k$ coefficients for the PDF of $Y_{11}^{(1)}$

**(c)** The $c_k$ coefficients for the PDF of $Y_{15}^{(2)}$

**(d)** The $c_k$ coefficients for the PDF of $Y_{20}^{(3)}$

**Figure 3.** The sequences of $c_k$'s in the Laguerre expansion (3) for the PDFs of $Y_n^{(j)}$, $j = 1, 2, 3$ set in Example 1, in which the values of parameters $\alpha_i^{(j)}$, $\nu_i^{(j)}$, and $\delta_i^{(j)}$ are displayed in Figure 1a–d.

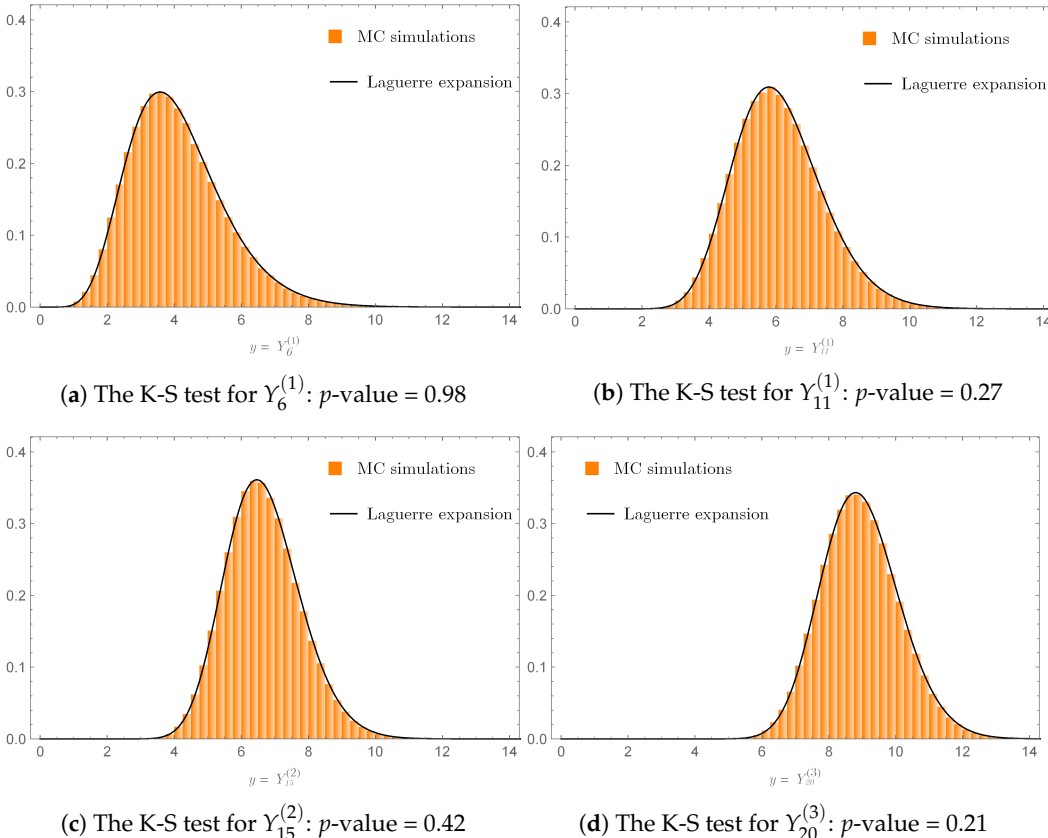

**(a)** The K-S test for $Y_6^{(1)}$: $p$-value = 0.98

**(b)** The K-S test for $Y_{11}^{(1)}$: $p$-value = 0.27

**(c)** The K-S test for $Y_{15}^{(2)}$: $p$-value = 0.42

**(d)** The K-S test for $Y_{20}^{(3)}$: $p$-value = 0.21

**Figure 4.** The PDFs of the $Y_n^{(j)}$'s in Example 1 computed by using the Laguerre expansion (3), against the corresponding histograms obtained from MC simulations, in which the $p$-values based on the K-S tests are greater than 0.1 (the significant level).

To extend our study to various cases of the linear combination (40) as introduced in Section *3.3*, with increasing values of $n$, we further considered the PDFs of $Y_{11}^{(1)} = \sum_{i=1}^{11} \alpha_i^{(1)} X_i^{(1)}$, $Y_{15}^{(2)} = \sum_{i=1}^{15} \alpha_i^{(2)} X_i^{(2)}$, and $Y_{20}^{(3)} = \sum_{i=1}^{20} \alpha_i^{(3)} X_i^{(3)}$. The values of the parameters $\alpha_i^{(j)}, \nu_i^{(j)}$, and $\delta_i^{(j)}$ for $j = 1, 2, 3$ computed by using (42)–(44) are plotted in Figure *1b–d*. By following the procedure as previously described for determining the PDF of $Y_6^{(1)}$, we thus obtained the sequences of $c_k$'s as shown in Figure *3b–d* along with the PDFs of $Y_{11}^{(1)}, Y_{15}^{(2)}$, and $Y_{20}^{(3)}$, denoted by $f_{Y_{11}^{(1)}}^{(\beta)}(y), f_{Y_{15}^{(2)}}^{(\beta)}(y)$, and $f_{Y_{20}^{(3)}}^{(\beta)}(y)$, as shown in Figure *4b–d*, respectively. It is clearly seen from Figure *4b–d* that the graphs of the PDFs markedly matched their corresponding histograms obtained from MC simulations.

Next, we illustrate the fitness as previously discussed by employing the Kolmogorov–Smirnov (K-S) test [42]. Figure *4a–d* also display the p-values for the K-S tests computed from 1000 random samples generated by the PDFs $f_{Y_6^{(1)}}^{(\beta)}(y), f_{Y_{11}^{(1)}}^{(\beta)}(y), f_{Y_{15}^{(2)}}^{(\beta)}(y)$, and $f_{Y_{20}^{(3)}}^{(\beta)}(y)$ and their corresponding random variables $Y_6^{(1)}, Y_{11}^{(1)}, Y_{15}^{(2)}$, and $Y_{20}^{(3)}$, respectively. As shown in Figure *4a–d*, the p-values obtained fell into the area of the acceptance region in which we set the significant levels for the K-S tests to be 0.1; all the resulting null hypotheses of the K-S tests were accepted. Therefore, the PDFs $f_{Y_6^{(1)}}^{(\beta)}(y), f_{Y_{11}^{(1)}}^{(\beta)}(y), f_{Y_{15}^{(2)}}^{(\beta)}(y)$, and $f_{Y_{20}^{(3)}}^{(\beta)}(y)$ computed by using the Laguerre expansion (3) appeared consistent with their corresponding histogram of random samples generated from $Y_6^{(1)}$, $Y_{11}^{(1)}, Y_{15}^{(2)}$, and $Y_{20}^{(3)}$, respectively.

*5.2. The Performance of Our Explicit Formula for $\mathbb{E}[Y_n^\gamma]$*

**Example 2.** *In our next example, we demonstrate the performance of our explicit Formula (8) by selecting $Y_{11}^{(1)}, Y_{15}^{(2)}$, and $Y_{20}^{(3)}$ given in Example 1 to be our case study.*

Firstly, we computed the values of $\mathbb{E}\left[\left(Y_{11}^{(1)}\right)^\gamma\right]$ for $\gamma \in [0, 2]$ by using our explicit Formula (8) with $K = K_1 = 100$. Next, we plotted these values against the results obtained from MC simulations with $n_p = 10, 20, 50, 100, 500, 2000$, which were the numbers of sample paths used in the MC simulations, in Figure *5a–f*, respectively. Next, we similarly applied this procedure to investigate the accuracy of our explicit Formula (8) for $\mathbb{E}\left[\left(Y_{15}^{(2)}\right)^\gamma\right]$ and $\mathbb{E}\left[\left(Y_{20}^{(3)}\right)^\gamma\right]$. The results obtained for $\mathbb{E}\left[\left(Y_{15}^{(2)}\right)^\gamma\right]$, where $\gamma = 2, 2.01, \ldots, 3$, with $K = K_2 = 300$ and $\mathbb{E}\left[\left(Y_{20}^{(3)}\right)^\gamma\right]$, where $\gamma = 3, 3.01, \ldots, 4$, with $K = K_3 = 400$, are displayed in Figure *6a,c,e* and Figure *6b,d,f*, respectively. Evidently, the variation of the approximate values from MC simulations decreased when $n_p$ increased for all chosen $\gamma$'s, demonstrating the convergence of the approximate values from MC simulations to the one computed by using our explicit Formula (8).

In order to verify our result presented in Theorem *3*, we used (11) to compute the truncation errors of $\mathbb{E}\left[\left(Y_{11}^{(1)}\right)^\gamma\right], \mathbb{E}\left[\left(Y_{15}^{(2)}\right)^\gamma\right]$, and $\mathbb{E}\left[\left(Y_{20}^{(3)}\right)^\gamma\right]$, denoted by $E_{k,\infty}^{(\gamma,1)}, E_{k,\infty}^{(\gamma,2)}$, and $E_{k,\infty}^{(\gamma,3)}$, respectively, when $\gamma = \frac{1}{2} \notin \mathbb{N}$ and $\gamma = 1, 2 \in \mathbb{N}$ for $k = 0, \ldots, 10$. The truncation errors obtained are tabulated in Table *1*. We clearly see from Table *1* that $E_{k,\infty}^{(\gamma,j)}$ tended to zero when $k$ increased for all $j = 1, 2, 3$ and selected $\gamma$. In particular, $E_{k,\infty}^{(\gamma,j)} = 0$ for $k \geq \gamma$ when $\gamma = 1, 2$. This confirmed our result presented in Theorem *4* that our explicit Formula (17) could be used to compute $\mathbb{E}[Y_n^\gamma]$ without producing truncation errors when $\gamma \in \mathbb{N}$; it should be remarked from Table *1* that the truncation errors could be very large when $K + 1$ was less than $\gamma$ for $\gamma \in \mathbb{N}$. Although utilizing our explicit Formula (8) for computing $\mathbb{E}[Y_n^\gamma]$ when $\gamma \in \mathbb{R}_+$ and $\gamma \notin \mathbb{N}$ always produced a truncation error, our result presented in Theorem *3* ensured that the truncation error tended to zero as $K$ increased.

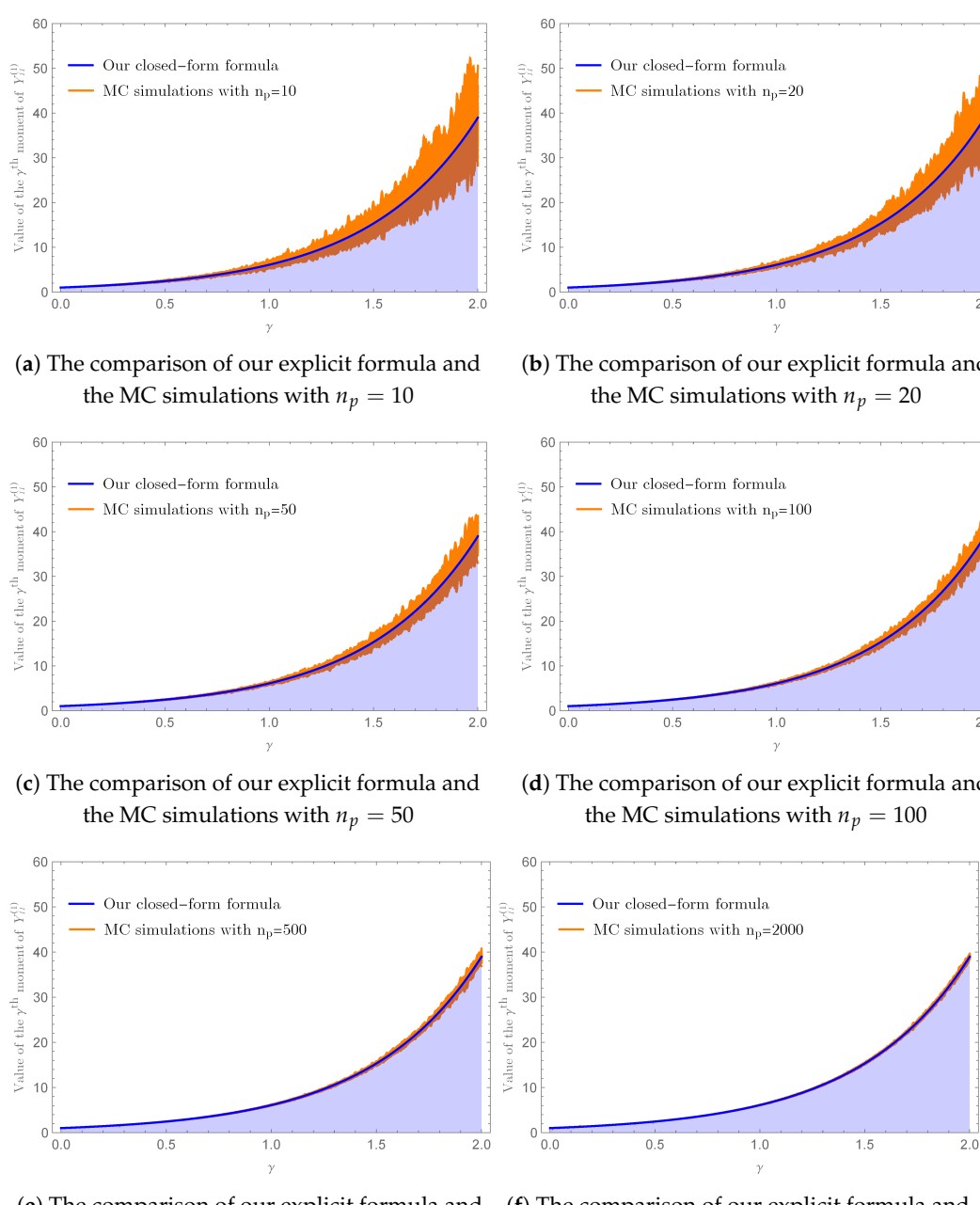

(**a**) The comparison of our explicit formula and the MC simulations with $n_p = 10$

(**b**) The comparison of our explicit formula and the MC simulations with $n_p = 20$

(**c**) The comparison of our explicit formula and the MC simulations with $n_p = 50$

(**d**) The comparison of our explicit formula and the MC simulations with $n_p = 100$

(**e**) The comparison of our explicit formula and the MC simulations with $n_p = 500$

(**f**) The comparison of our explicit formula and the MC simulations with $n_p = 2000$

**Figure 5.** The variation on the approximate values of $\mathbb{E}\left[(Y_{11}^{(1)})^\gamma\right]$ obtained from MC simulations in Example 2 with an increasing number of sample paths, demonstrating the convergence of the approximate values obtained from MC simulations to the one computed by using our Formula (8) with $K = K_1 = 100$ terms in the infinite series, when the number of sample paths approaches infinity.

We finish this example by illustrating the efficiency of our explicit Formula (8) over the MC simulations. As shown in Figures 5a–f and 6a–f, we needed to increase the value of $n_p$ in the MC simulations in order to reduce the variations on the approximate values of $\mathbb{E}\left[\left(Y_{11}^{(1)}\right)^\gamma\right]$, $\mathbb{E}\left[\left(Y_{15}^{(2)}\right)^\gamma\right]$, and $\mathbb{E}\left[\left(Y_{20}^{(3)}\right)^\gamma\right]$, respectively, which could be time-consuming. For example, to yield an absolute difference between the exact value of $\mathbb{E}\left[\left(Y_{20}^{(3)}\right)^2\right]$ computed from our explicit Formula (8) and its approximate value obtained from MC simulations to be less than $10^{-3}$, our numerical experiment required MC simulations with $n_p = 10^7$ consuming 19 s, while implementing our explicit Formula (8) with $K = 3$ took just 0.001 s.

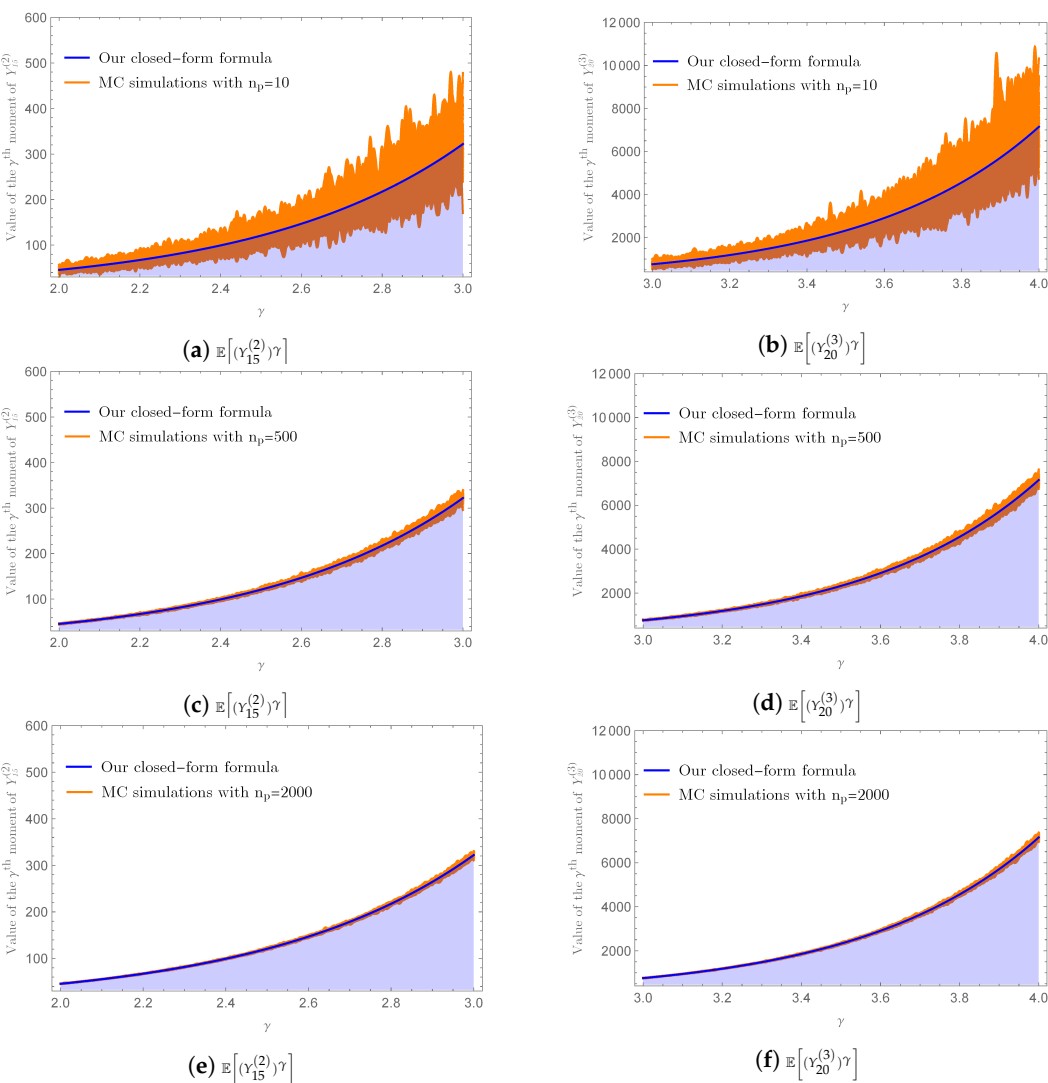

**Figure 6.** The variations on the approximate values of $\mathbb{E}\left[(Y_{15}^{(2)})^\gamma\right]$ and $\mathbb{E}\left[(Y_{20}^{(3)})^\gamma\right]$ obtained from MC simulations in Example 2 with an increasing number of sample paths, demonstrating the convergence of the approximate values obtained from MC simulations to the ones computed by using our Formula (8) with $K = K_1 = 300$ and $K = K_2 = 400$ terms in the infinite series, respectively, when the number of sample paths approaches infinity.

**Table 1.** The truncation errors of $\mathbb{E}\left[\left(Y_{11}^{(1)}\right)^\gamma\right]$, $\mathbb{E}\left[\left(Y_{15}^{(2)}\right)^\gamma\right]$, and $\mathbb{E}\left[\left(Y_{20}^{(3)}\right)^\gamma\right]$, denoted by $E_{k,\infty}^{(\gamma,1)}$, $E_{k,\infty}^{(\gamma,2)}$, and $E_{k,\infty}^{(\gamma,3)}$, respectively, computed in Example 2 by using (11) when $\gamma = \frac{1}{2} \notin \mathbb{N}$ and $\gamma = 1, 2 \in \mathbb{N}$ for $k = 0, \ldots, 10$.

| $k$ | $\gamma = 1/2$ | | | $\gamma = 1$ | | | $\gamma = 2$ | | |
|---|---|---|---|---|---|---|---|---|---|
| | $E_{k,\infty}^{(\gamma,1)}$ | $E_{k,\infty}^{(\gamma,2)}$ | $E_{k,\infty}^{(\gamma,3)}$ | $E_{k,\infty}^{(\gamma,1)}$ | $E_{k,\infty}^{(\gamma,2)}$ | $E_{k,\infty}^{(\gamma,3)}$ | $E_{k,\infty}^{(\gamma,1)}$ | $E_{k,\infty}^{(\gamma,2)}$ | $E_{k,\infty}^{(\gamma,3)}$ |
| 0 | $6.2 \times 10^{-1}$ | $1.2 \times 10^{0}$ | $1.6 \times 10^{0}$ | $3.5 \times 10^{0}$ | $7.6 \times 10^{0}$ | $1.2 \times 10^{1}$ | $5.6 \times 10^{1}$ | $1.6 \times 10^{2}$ | $3.8 \times 10^{2}$ |
| 1 | $6.5 \times 10^{-2}$ | $1.9 \times 10^{-1}$ | $2.8 \times 10^{-1}$ | 0 | 0 | 0 | $1.3 \times 10^{1}$ | $5.8 \times 10^{1}$ | $1.5 \times 10^{2}$ |
| 2 | $1.2 \times 10^{-2}$ | $5.5 \times 10^{-2}$ | $9.1 \times 10^{-2}$ | 0 | 0 | 0 | 0 | 0 | 0 |
| 3 | $3.2 \times 10^{-3}$ | $2.0 \times 10^{-2}$ | $3.5 \times 10^{-2}$ | 0 | 0 | 0 | 0 | 0 | 0 |
| 4 | $9.1 \times 10^{-4}$ | $7.7 \times 10^{-3}$ | $1.5 \times 10^{-2}$ | 0 | 0 | 0 | 0 | 0 | 0 |
| 5 | $2.8 \times 10^{-4}$ | $3.2 \times 10^{-3}$ | $6.8 \times 10^{-3}$ | 0 | 0 | 0 | 0 | 0 | 0 |
| 6 | $9.2 \times 10^{-5}$ | $1.4 \times 10^{-3}$ | $3.2 \times 10^{-3}$ | 0 | 0 | 0 | 0 | 0 | 0 |
| 7 | $3.1 \times 10^{-5}$ | $6.3 \times 10^{-4}$ | $1.5 \times 10^{-3}$ | 0 | 0 | 0 | 0 | 0 | 0 |
| 8 | $1.1 \times 10^{-5}$ | $2.9 \times 10^{-4}$ | $7.7 \times 10^{-4}$ | 0 | 0 | 0 | 0 | 0 | 0 |
| 9 | $4.1 \times 10^{-6}$ | $1.3 \times 10^{-4}$ | $3.9 \times 10^{-4}$ | 0 | 0 | 0 | 0 | 0 | 0 |
| 10 | $1.6 \times 10^{-6}$ | $6.7 \times 10^{-5}$ | $1.9 \times 10^{-4}$ | 0 | 0 | 0 | 0 | 0 | 0 |

### 5.3. Extended Results for the ECIR Process with Time-Varying Dimension

We next shift our attention to our explicit formulas for the TPDF and the moments of ECIR processes with time-varying dimension. We specifically set the parameter functions of the ECIR process $V_t$ described by (19) with time-varying dimension $\mathbf{d}(t)$ defined in (20) as follows:

$$\kappa(t) = 0.1 + 0.2t + 0.3e^{\cos^2(t+2)}, \tag{45}$$

$$\theta(t) = 0.1 + 0.5e^{2\sin(t+2)}, \tag{46}$$

and

$$\sigma(t) = (0.4 + 0.1t)e^{\sin(t+2)} \tag{47}$$

for $t \in [0,3]$.

Figure 7a–f display variations of the three parameter functions (45)–(47) as well as $\mathbf{d}(t)$, $\mathbf{d}^{(1)}(t)$, and $\tau(t,0)$ defined in (21), respectively, as given in the figure. It should be noticed from Figure 7d that Assumption 1 was fulfilled on $[0,3]$, while Assumption 2 was violated, e.g., $\mathbf{d}^{(1)}(1.5) < 0$, as shown in Figure 7e. Consequently, we set the time domain for this study to be $\mathbf{D}_1 := [0,1]$, ensuring that both Assumptions 1 and 2 were satisfied.

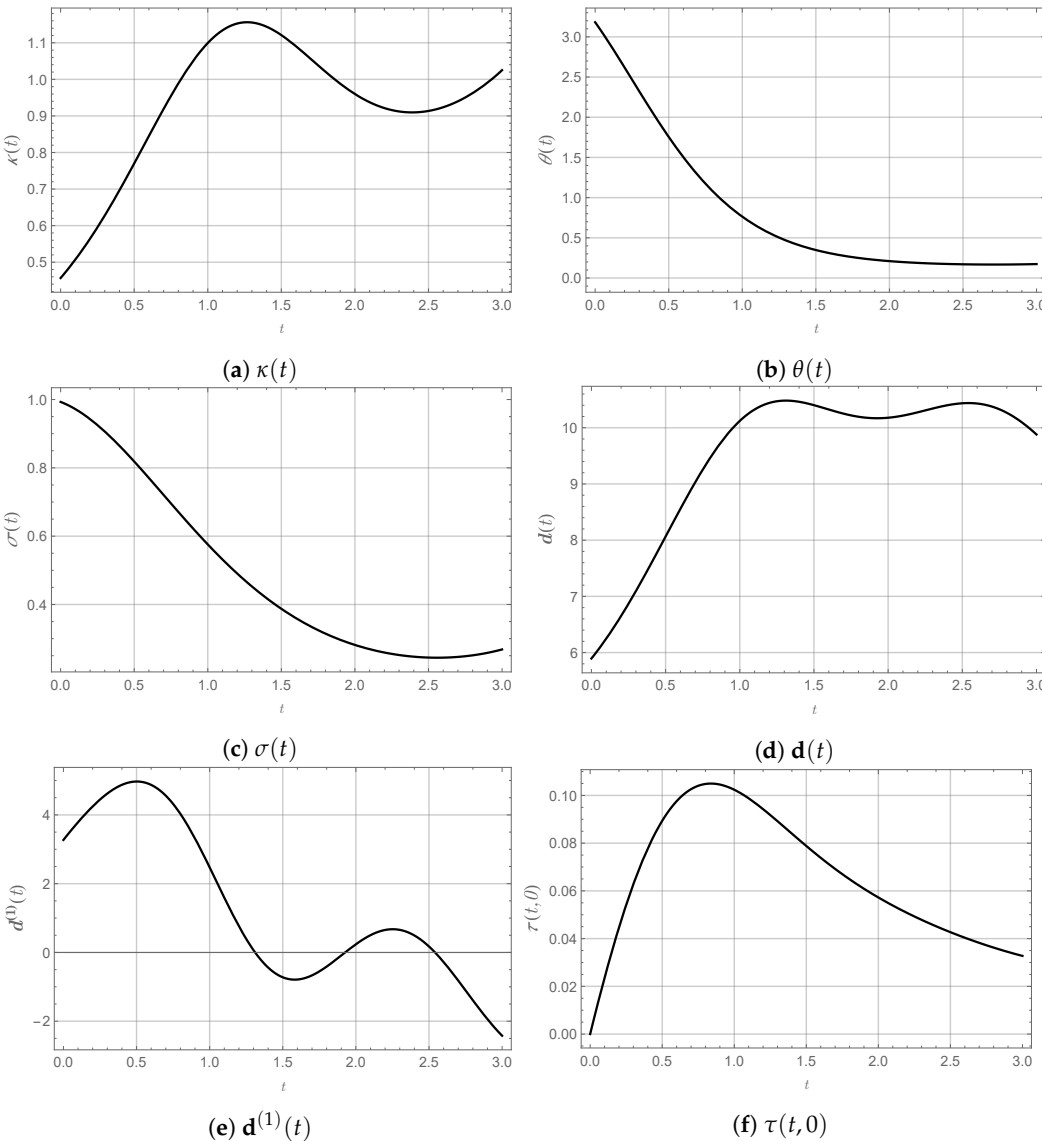

**Figure 7.** Variations of the three parameter functions (45)–(47) as well as $\mathbf{d}(t)$, $\mathbf{d}^{(1)}(t)$, and $\tau(t,0)$ for $t \in [0,3]$ in Example 3.

5.3.1. The Accuracy of Our Explicit Formula for the TPDF of the ECIR Process with Time-Varying Dimension

**Example 3.** *This example aimed to investigate the accuracy of our explicit Formula (31) for calculating the TPDF of $V_t$. We started by implementing the result presented in Theorem 10. Consider the convergence of the PDF of $\hat{Y}_n(t, v_0) := \sum_{i=1}^{n} \hat{\alpha}_i \hat{X}_i$ to the PDF of $V_t|v_0$ as $n$ approaches infinity, written in (22). We set $v_0 = 1, 2$, and $t = 0.1, 0.5, 1 \in \mathbf{D}_1$. Then, we computed the approximate PDFs of $\hat{Y}_n(t, v_0)$ for $n = 1, 2, 3$, based on the random samples drawn from populations distributed according to noncentral chi-square and chi-square distributions with a number of sample $n_p = 10^6$. On the other hand, as a benchmark, the approximate PDFs of $V_t|v_0$ for all $v_0 = 1, 2$, and $t = 0.1, 0.5, 1$ were obtained by using the sample paths generated from (19) as follows.*

*Figure 8a–f show that the PDF of $\hat{Y}_n(t, v_0)$ tended to align better with the histogram of $V_t|v_0$ representing the PDF of $V_t|v_0$ when $n$ increased for all $v_0 = 1, 2$, and $t = 0.1, 0.5, 1$, demonstrating $\hat{Y}_n(t, v_0)$ converged in distribution to $V_t|v_0$ as $n$ approached infinity. However, as shown in Figure 8a–f, the number of terms used in the summation $\sum_{i=1}^{n} \hat{\alpha}_i \hat{X}_i$ had to be increased in order to obtain a better approximation for the PDF of $V_t|v_0$. This is a major drawback of implementing (22), which requires the exact PDFs of $\hat{Y}_n(t, v_0)$ and $V_t|v_0$ in order to estimate errors occurring for all $n$.*

*The problem mentioned above can completely be solved by employing our explicit Formula (31) for obtaining the exact PDF of $V_t|v_0$. Truncation errors occurring when implementing the infinite series in (31) can be estimated by applying Lemma 1. To demonstrate the accuracy of our explicit Formula (31), we computed $f_{V_t}(v, t|v_0)$ for $v_0 = 1, 2$, $t = 0.1, 0.5, 1$ and $v > 0$ by setting $K = 20$. Sequences of $|\hat{c}_k(t, v_0)|$'s computed from (32) and (33) for $v_0 = 1, 2$ and $t = 0.1, 0.5, 1$ are displayed in Figure 9a,b, showing that the coefficients tended to zero when $k$ increased. This ensured the truncation errors vanished when $K$ approached infinity.*

*Figure 8a–f also display the graphs of $f_{V_t}(v, t|v_0)$ against the corresponding histograms of $V_t|v_0$ for $t = 0.1, 0.5, 1 \in \mathbf{D}_1$ and $v_0 = 1, 2$, obtained from the sample paths generated from (19). It is readily seen that the graphs of $f_{V_t}(v, t|v_0)$ gracefully matched the corresponding histograms. Following the K-S tests with the significance level of 10% employed in Example 1 to determine the equivalence between two distributions, the minimum of the p-values was 0.14, implying that there was no significance difference between $f_{V_t}(v, t|v_0)$ obtained from our explicit Formula (31) and the corresponding histograms of the random samples generated from (19).*

*The validity of our explicit Formula (31) admittedly becomes questionable when Assumption 2 is violated. To investigate, we extended the time domain from $\mathbf{D}_1 := [0, 1]$ to $\mathbf{D}_2 := [0, 3]$ and considered $f_{V_t}(v, t|v_0)$ for $v > 0$, $v_0 = 1, 2$, and $t = 0.03, 0.75, 1.5, 2.25, 3 \in \mathbf{D}_2$. Figure 7e visually shows that at $t = 1.5, 3$, $\mathbf{d}^{(1)}(1.5)$ and $\mathbf{d}^{(1)}(3)$ were negative and hence Assumption 2 was violated, but $f_{V_t}(v, t|v_0)$ was finite for all $v > 0$, $v_0 = 1, 2$, and $t \in \mathbf{D}_2$, as shown in Figure 10a,b.*

*Moreover, the graph of $f_{V_t}(v, t|v_0)$ appears to match the corresponding histograms of the random samples from (19) based on MC simulations because the minimum of the p-values from the K-S tests was 20%. The results obtained suggested that the condition in Assumption 2 could perhaps be replaced with $\mathbf{d}^{(1)}(t) < \infty$ for all $t \in [0, T]$ when our explicit Formula (31) was employed for computing the TPDF of the ECIR process (19). That being said, Assumption 2 remains an important ingredient of our analysis since the degrees of freedom $\hat{v}_i$ given in (27) are not allowed to be negative from the definitions of noncentral chi-square and chi-square distributions.*

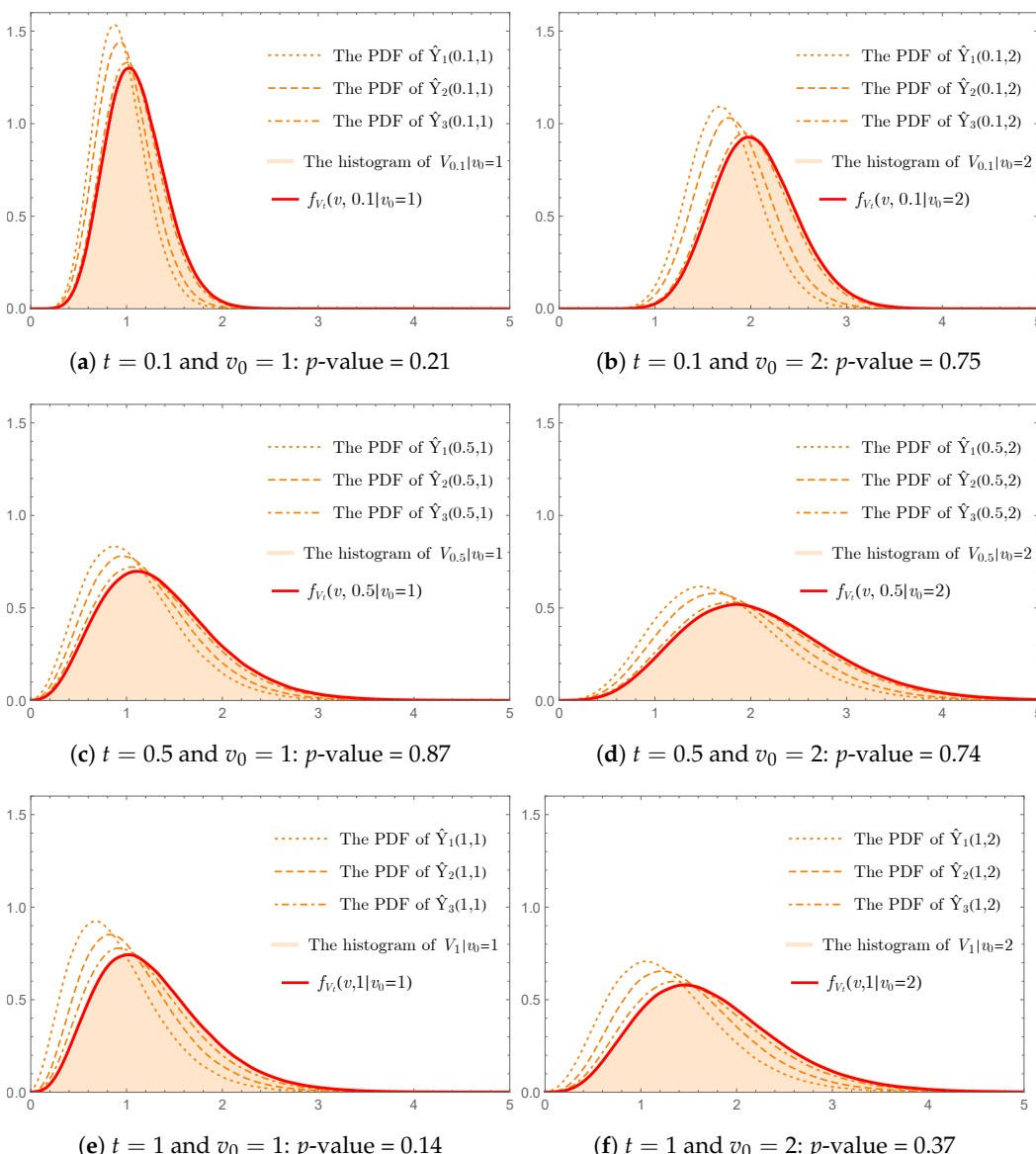

**Figure 8.** The convergence of $\hat{Y}_n(t, v_0) = \sum_{i=1}^{n} \hat{\alpha}_i \hat{X}_i$ in distribution to $V_t|v_0$ as $n$ approaches infinity tested by using MC simulations in Example 3, demonstrating the result (22) presented in Theorem 10, against $f_{V_t}(v, t|v_0)$ computed by using our explicit Formula (31).

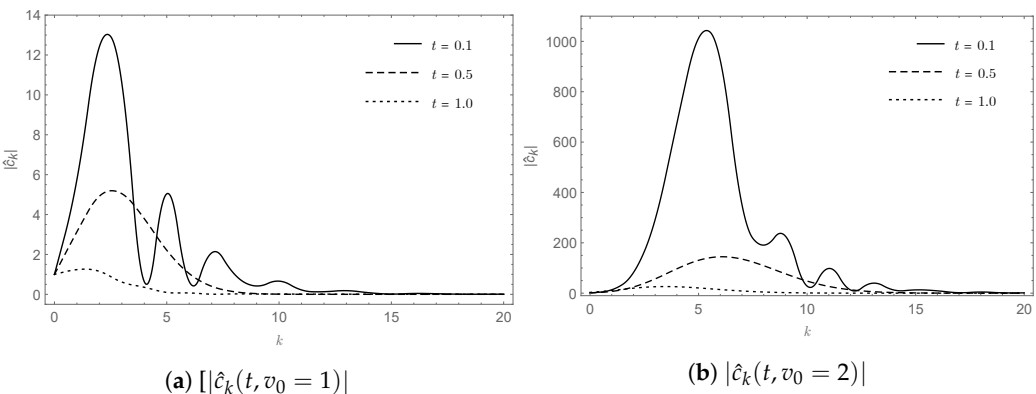

**Figure 9.** The sequences of $|\hat{c}_k(t, v_0)|$'s computed from (32) and (33) in Example 3.

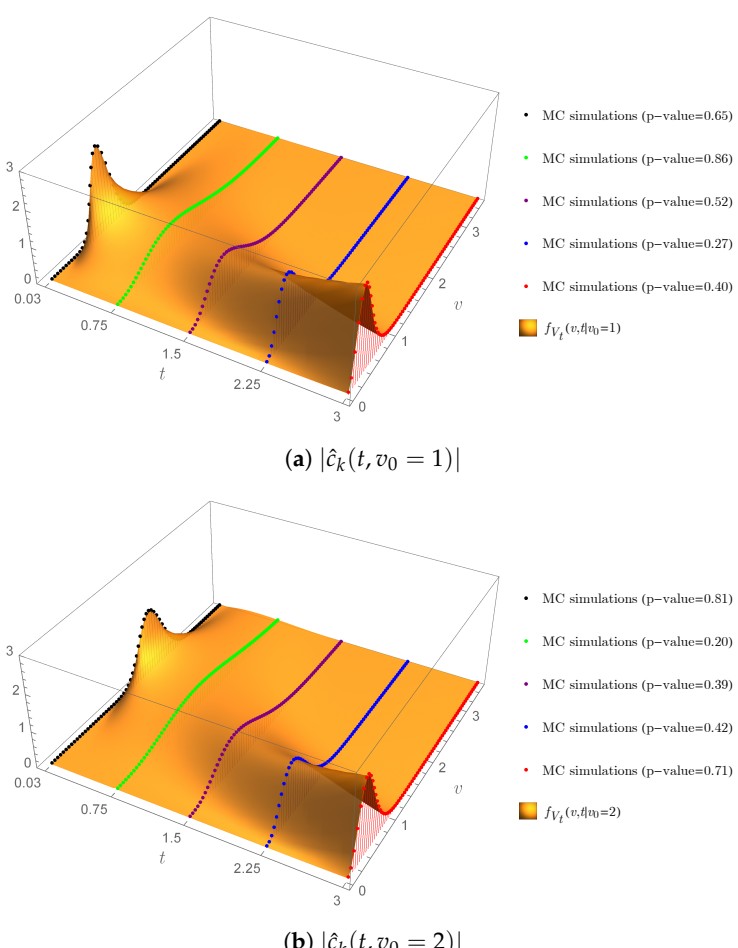

**(a)** $|\hat{c}_k(t, v_0 = 1)|$

**(b)** $|\hat{c}_k(t, v_0 = 2)|$

**Figure 10.** The graphs of $f_{V_t}(v, t|v_0)$ computed from our explicit Formula (31), against the corresponding histograms of the random samples generated from (19) based on MC simulations for $t = 0.03, 0.75, 1.5, 2.25, 3$, along with the *p*-values in parentheses obtained from the K-S tests with a significant level of 0.1 in Example 3.

5.3.2. The performance of Our Explicit Formula for $U_E^{(\gamma)}(t|v_0)$

**Example 4.** *In our last example, we illustrate the accuracy and efficiency of our explicit Formula (38) for computing the $\gamma^{th}$ conditional moment of the ECIR process (19) with time-varying dimension. As previously discussed in Section 4.2.1, we demonstrate the advantages of using our explicit Formula (38) over the explicit formula presented in Theorem 2.1 of Rujivan [12] by setting the parameter functions of the ECIR process (19) to follow (45)–(47) as used in Example 3.*

*Firstly, we considered the accuracy of our explicit Formula (38) and the one written in Equation (2.2) of Rujivan [12] by setting $\gamma = 0.5, 1, 1.5, 2, 2.5, 3$. Let $v_0 = 1$ and $t = 1$. For each $\gamma$, we computed two sequences of $U_E^{(\gamma, K)}(t|v_0)$ for $K = 0, \ldots, 5$, from our explicit formulas (38) and the one written in Equation (2.2) of Rujivan [12] with the number of terms, $K + 1$, used in the infinite series on the RHS of (38) and Equation (2.2) of Rujivan [12]. Figure 11a–f display the graphs of the two sequences of $U_E^{(\gamma, K)}(t|v_0)$ against the approximate value of $U_E^{(\gamma)}(t|v_0)$ from MC simulations based on the ECIR process (19), demonstrating that the two sequences of $U_E^{(\gamma, K)}(t|v_0)$ converged to the corresponding approximate value of $U_E^{(\gamma)}(t|v_0)$ obtained from MC simulations when K increased.*

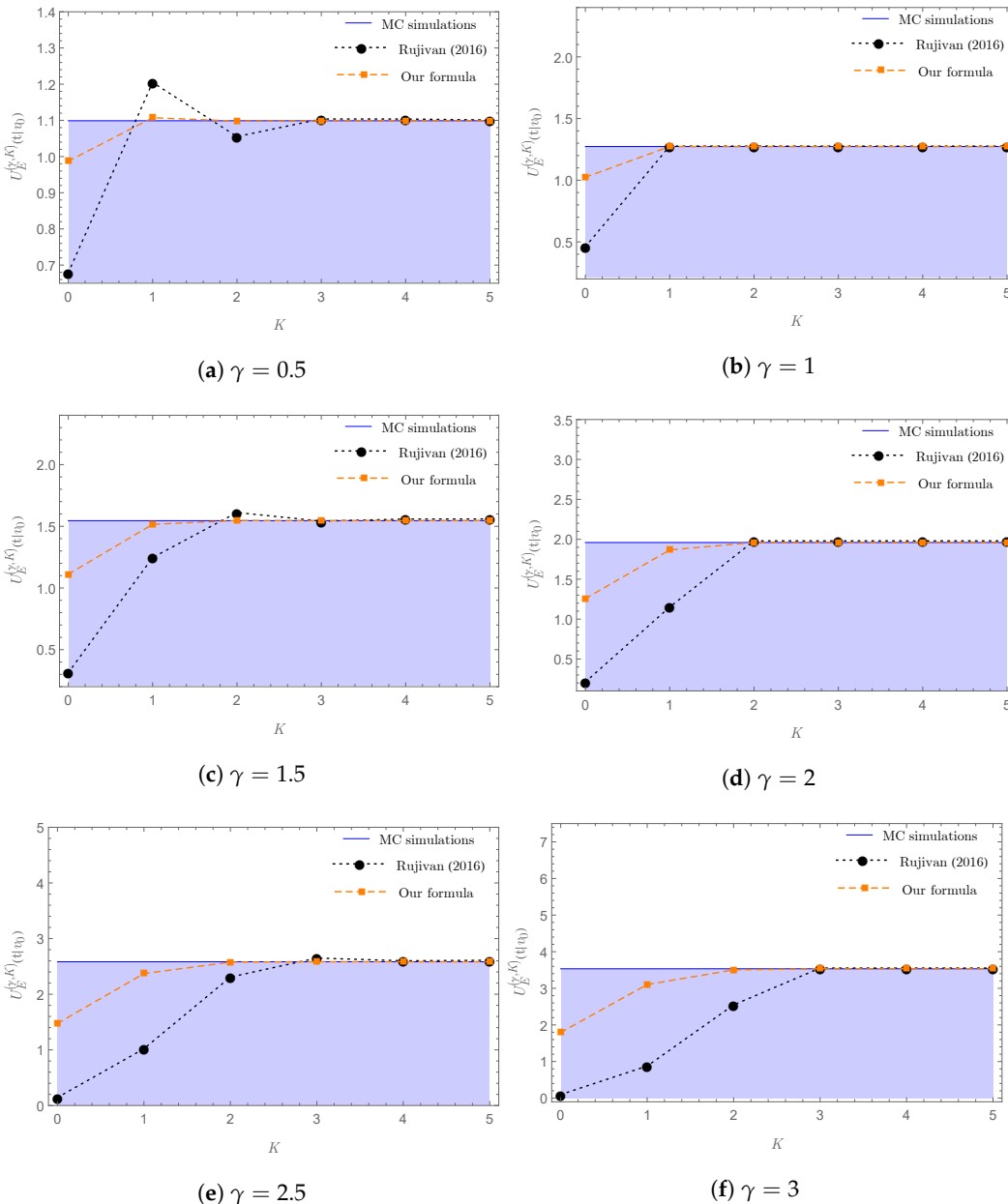

(**a**) $\gamma = 0.5$

(**b**) $\gamma = 1$

(**c**) $\gamma = 1.5$

(**d**) $\gamma = 2$

(**e**) $\gamma = 2.5$

(**f**) $\gamma = 3$

**Figure 11.** The sequences of $U_E^{(\gamma,K)}(t|v_0)$ for $K = 0,\dots,5$, computed from our explicit Formula (38) and the explicit formula written in Equation (2.2) of Rujivan [12], against the approximate value of $U_E^{(\gamma)}(t|v_0)$ obtained from MC simulations based on the ECIR process (19) in Example 4 by setting $v_0 = 1$ and $t = 1$.

It should also be pointed out from Figure 11a–f that the sequence of $U_E^{(\gamma,K)}(t|v_0)$ computed from (38) converged to the corresponding approximate value of $U_E^{(\gamma)}(t|v_0)$ obtained from the MC simulations faster than the sequence of $U_E^{(\gamma,K)}(t|v_0)$ computed from Equation (2.2) of Rujivan [12]. Moreover, as shown in Figure 11b,d,f, the two sequences of $U_E^{(\gamma,K)}(t|v_0)$ coincided for $K + 1 \geq m$ when $\gamma = m$ was a positive integer, demonstrating the consistency of our closed-from Formula (39) and the explicit formula written in Equation (2.13) in Theorem 2.2 of Rujivan [12].

Secondly, we investigated the efficiency of our explicit Formula (38) over the one written in Equation (2.2) of Rujivan [12] for calculating $U_E^{(\gamma)}(t|v_0)$ by choosing $\gamma = 0.5$ in our case study. Figure 12a,b illustrate the computational times (seconds) used to compute $U_E^{(\gamma,K)}(t|v_0)$ from the two explicit formulas and the reduction (in folds) of computational time for $K = 1,\dots,10$, respectively.

*As expected, implementing our explicit Formula (38) consumed considerably less time and effort than implementing the explicit formula written in Equation (2.2) of Rujivan [12], in particular, the reduction was more than sixfold when $K \geq 10$.*

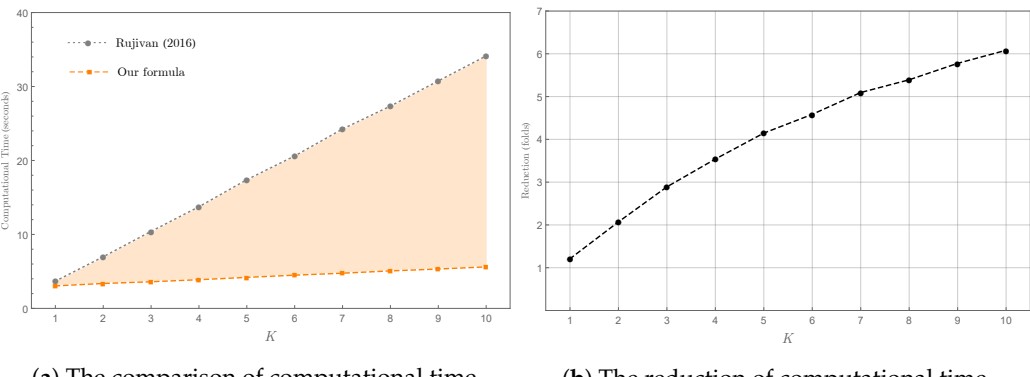

(**a**) The comparison of computational time        (**b**) The reduction of computational time

**Figure 12.** The efficiency of our explicit Formula (38) over the explicit formula written in Equation (2.2) of Rujivan [12] for calculating $U_E^{(\gamma)}(t|v_0)$ in Example 4 by setting $\gamma = 0.5$, $v_0 = 1$, and $t = 1$.

## 6. Conclusions

In this paper, we presented the first explicit formula expressed in terms of generalized hypergeometric functions for computing the $\gamma^{\text{th}}$ moment of a conic combination of $n$ independent noncentral chi-square random variables defined in (1), when the conic coefficients were not all identical for any integer $n \geq 2$ and real number $\gamma \in \mathbb{R}_+$. Moreover, the truncation errors occurring by implementing our explicit formulas were determined analytically. We extended our result to various types of random variables which were independent and could be transformed to noncentral chi-square random variables. For validation purposes, several numerical examples were presented to show the performance of our explicit formulas compared with MC simulations. Furthermore, we highlighted an interesting application of our explicit formulas in interest rate modeling by expressing the exact TPDF of the ECIR process with time-varying dimension in terms of generalized Laguerre functions. As a result, a novel explicit formula for the $\gamma^{\text{th}}$ conditional moment of the ECIR process was obtained and tested, and we concluded that the distinguished feature of our current analytical approach lay in its computational efficiency, which was superior to that of the other existing methods from the literature.

**Author Contributions:** Conceptualization, S.R., A.S., K.C. and N.R.; methodology, S.R. and A.S.; software, S.R. and A.S.; validation, S.R., A.S., K.C. and N.R.; formal analysis, S.R., A.S., K.C. and N.R.; investigation, S.R., A.S., K.C. and N.R.; writing—original draft preparation, S.R. and A.S.; writing—review and editing, S.R., K.C. and N.R.; visualization, S.R. and A.S.; supervision, K.C. and N.R.; project administration, S.R. and K.C.; funding acquisition, S.R. All authors have read and agreed to the published version of the manuscript.

**Funding:** This research project was supported by the National Research Council of Thailand (NRCT):NRCT5-RGJ63016-150 and Walailak University partial funding contract no. 01/2562 for the first and the second authors. Furthermore, the first author received funding support from the NSRF, Thailand, via the Program Management Unit for Human Resources and Institutional Development, Research and Innovation (grant no. B05F640202).

**Institutional Review Board Statement:** Not applicable.

**Informed Consent Statement:** Not applicable.

**Data Availability Statement:** Not applicable.

**Acknowledgments:** We are grateful for the suggestions from the anonymous referees that have substantially improved the quality and presentation of the results. All errors are the authors' own responsibility.

**Conflicts of Interest:** The authors declare no conflict of interest.

## Abbreviations

The following abbreviations are used in this manuscript:

| | |
|---|---|
| ECIR | Extended Cox–Ingersoll–Ross |
| CIR | Cox–Ingersoll–Ross |
| K-S | Kolmogorov–Smirnov |
| MC | Monte Carlo |
| ODE | Ordinary differential equation |
| PDE | Partial differential equation |
| PDF | Probability density function |
| SDE | Stochastic differential equation |
| TPDF | Transition probability density function |

## Appendix A. Omitted Proofs from Section 2

**Proof of Theorem 1.** To obtain the PDF of $Y_n$ as written in (3), we use the results proposed in Section 3 of [16] as follows. From (1), we let the random variable $Q_n$ defined in [16] to be $Y_n$ and consider the coefficients $\alpha_i > 0$ and random variables $X_i \sim \chi^2_{\nu_i}(\delta_i)$ for $i = 1, \ldots, n$. Then, we set $p = \mu_0 = \frac{\nu}{2}$ in Equations (3.2), (3.4a), and (3.4b) of [16]. As a result, we have $c_0 = 1$. Moreover, the formulas of the remaining $c_k$ coefficients on the RHS of Equation (3.2), as written in Equations (3.4a) and (3.4b), reduce to (5) and (6)–(7), respectively. □

## Appendix B. Omitted Proofs from Section 3

**Proof of Theorem 2.** We set

$$C_{k,\gamma}(y) := \frac{1}{(2\beta)^{\frac{\nu}{2}}} \frac{k!}{\Gamma(\frac{\nu}{2}+k)} c_k \mathbf{L}_k^{\left(\frac{\nu}{2}-1\right)}\left(\frac{y}{2\beta}\right) e^{-\frac{y}{2\beta}} y^{\frac{\nu}{2}-1+\gamma} \tag{A1}$$

for $\gamma \in \mathbb{R}_+$ and $k = 0, 1, \ldots$, where the $c_k$'s, $\nu$, and $\beta$ are given in Theorem 1.

Observe that

$$\begin{aligned}
\int_0^\infty C_{k,\gamma}(y)dy &= \frac{k! c_k}{(2\beta)^{\frac{\nu}{2}}\Gamma(\frac{\nu}{2}+k)} \int_0^\infty \mathbf{L}_k^{\left(\frac{\nu}{2}-1\right)}\left(\frac{y}{2\beta}\right) e^{-\frac{y}{2\beta}} y^{\frac{\nu}{2}-1+\gamma} dy \\
&= \frac{k!(2\beta)^\gamma c_k}{\Gamma(\frac{\nu}{2}+k)} \int_0^\infty \mathbf{L}_k^{\left(\frac{\nu}{2}-1\right)}(y) e^{-y} y^{\frac{\nu}{2}-1+\gamma} dy \\
&= \frac{(2\beta)^\gamma c_k \Gamma\left(\frac{\nu}{2}+\gamma\right)}{\Gamma\left(\frac{\nu}{2}\right)} {}_2\mathbf{F}_1\left(-k, \frac{\nu}{2}+\gamma; \frac{\nu}{2}; 1\right) \\
&= \frac{(2\beta)^\gamma c_k \Gamma\left(\frac{\nu}{2}+\gamma\right)}{\Gamma\left(\frac{\nu}{2}\right)} \frac{(-\gamma)_k}{\left(\frac{\nu}{2}\right)_k},
\end{aligned}$$

where the third equality follows from [43] and $(\cdot)_k$ denotes the usual Pochhammer symbol. By substituting the following identities to the above derivation

$$\Gamma\left(\frac{\nu}{2}\right) = \frac{\Gamma\left(\frac{\nu}{2}+k\right)}{\left(\frac{\nu}{2}\right)_k} \quad \text{and} \quad \Gamma\left(\frac{\nu}{2}+\gamma\right) = \frac{\Gamma\left(\frac{\nu}{2}+\gamma+k\right)}{\left(\frac{\nu}{2}+\gamma\right)_k}$$

we further obtain

$$
\begin{aligned}
\int_0^\infty C_{k,\gamma}(y)dy &= (2\beta)^\gamma c_k \frac{\Gamma\left(\gamma + k + \frac{\nu}{2}\right)}{\Gamma\left(k + \frac{\nu}{2}\right)} \frac{(-\gamma)_k}{\left(\frac{\nu}{2} + \gamma\right)_k} \\
&= (2\beta)^\gamma (-1)^k c_k \frac{\Gamma\left(\gamma + k + \frac{\nu}{2}\right)}{\Gamma\left(k + \frac{\nu}{2}\right)} \frac{(-\gamma)_k}{\left(1 - k - \frac{\nu}{2} - \gamma\right)_k} \\
&= (2\beta)^\gamma (-1)^k \frac{\Gamma\left(\gamma + k + \frac{\nu}{2}\right)}{\Gamma\left(k + \frac{\nu}{2}\right)} {}_2\mathbf{F}_1\left(-k, 1 - k - \frac{\nu}{2}; 1 - k - \frac{\nu}{2} - \gamma; 1\right) c_k,
\end{aligned}
\tag{A2}
$$

which is manifestly finite for all $\gamma \in \mathbb{R}_+$ and $k \in \mathbb{N} \cup \{0\}$.

The uniformly convergent series of $f_{Y_n}$ derived in (3) implies that the series $\sum_{k=0}^\infty C_{k,\gamma}(y)$ converges uniformly to $y^\gamma f_{Y_n}(y)$, i.e.,

$$
y^\gamma f_{Y_n}(y) = \sum_{k=0}^\infty C_{k,\gamma}(y).
\tag{A3}
$$

Applying (A2) and (A3) yields

$$
\begin{aligned}
\mathbb{E}[Y_n^\gamma] &= \int_0^\infty y^\gamma f_{Y_n}(y)dy \\
&= \sum_{k=0}^\infty \int_0^\infty C_{k,\gamma}(y)dy \\
&= (2\beta)^\gamma \sum_{k=0}^\infty (-1)^k \frac{\Gamma\left(\gamma + k + \frac{\nu}{2}\right)}{\Gamma\left(k + \frac{\nu}{2}\right)} {}_2\mathbf{F}_1\left(-k, 1 - k - \frac{\nu}{2}; 1 - k - \frac{\nu}{2} - \gamma; 1\right) c_k,
\end{aligned}
\tag{A4}
$$

and this finishes the proof. $\square$

**Proof of Corollary 1.** Firstly, we apply the Gauss summation formula [31] to the generalized hypergeometric functions in the infinite series on the RHS of (8) with $\gamma = \frac{1}{2}$ as follows:

$$
{}_2\mathbf{F}_1\left(-k, 1 - k - \frac{\nu}{2}; 1 - k - \frac{\nu}{2} - \frac{1}{2}; 1\right) = \frac{\Gamma\left(1 - k - \frac{\nu}{2} - \frac{1}{2}\right)\Gamma\left(k - \frac{1}{2}\right)}{\Gamma\left(1 - \frac{\nu}{2} - \frac{1}{2}\right)\Gamma\left(-\frac{1}{2}\right)}.
\tag{A5}
$$

Let $z_1 = k + \frac{\nu}{2}$ and $z_2 = \frac{\nu}{2}$. We use the properties of gamma functions to obtain the following relations:

$$
\Gamma\left(1 - k - \frac{\nu}{2} - \frac{1}{2}\right) = \Gamma\left(\frac{1}{2} - z_1\right) = \frac{(-4)^{z_1} z_1! \sqrt{\pi}}{(2z_1)!},
\tag{A6}
$$

$$
\Gamma\left(\frac{1}{2} + k + \frac{\nu}{2}\right) = \Gamma\left(\frac{1}{2} + z_1\right) = \frac{(2z_1)! \sqrt{\pi}}{(4)^{z_1} z_1!},
\tag{A7}
$$

$$
\Gamma\left(1 - \frac{\nu}{2} - \frac{1}{2}\right) = \Gamma\left(\frac{1}{2} - z_2\right) = \frac{(-4)^{z_2} z_2! \sqrt{\pi}}{(2z_2)!},
\tag{A8}
$$

$$
\Gamma\left(\frac{\nu + 1}{2}\right) = \Gamma\left(\frac{1}{2} + z_2\right) = \frac{(2z_2)! \sqrt{\pi}}{(4)^{z_2} z_2!}.
\tag{A9}
$$

Applying (A5)–(A9) to the coefficients of the infinite series on the RHS of (8) with $\gamma = \frac{1}{2}$ and simplifying the result obtained by using the Gauss summation formula yield

$$
(-1)^k \frac{\Gamma\left(\frac{1}{2} + k + \frac{\nu}{2}\right)}{\Gamma\left(k + \frac{\nu}{2}\right)} {}_2\mathbf{F}_1\left(-k, 1 - k - \frac{\nu}{2}; 1 - k - \frac{\nu}{2} - \frac{1}{2}; 1\right) = \frac{\Gamma\left(\frac{\nu+1}{2}\right)}{\Gamma\left(\frac{\nu}{2}\right)} {}_2\mathbf{F}_1\left(-k, \frac{\nu+1}{2}; \frac{\nu}{2}; 1\right)
\tag{A10}
$$

where

$$_2\mathbf{F}_1\left(-k, \frac{\nu+1}{2}; \frac{\nu}{2}; 1\right) = \frac{\Gamma\left(\frac{\nu}{2}\right)\Gamma\left(k - \frac{1}{2}\right)}{\Gamma\left(k + \frac{\nu}{2}\right)\Gamma\left(-\frac{1}{2}\right)} \tag{A11}$$

and this completes the proof. $\square$

**Proof of Corollary 2.** From (6) and (7), when $\beta = \alpha_i = 1$ for all $i = 1, \ldots, n$, and $\delta = \sum_{i=1}^{n} \delta_i > 0$, we have $d_1 = -\frac{1}{2}\delta$ and $d_j = 0$ for all $j = 2, 3, \ldots$ Applying the result obtained to (5) yields

$$c_k = \frac{(-1)^k \delta^k}{2^k k!} \tag{A12}$$

for $k = 1, 2, \ldots$ We set $n = 1$ and $X = Y_1$. Replacing the coefficients in the infinite series on the RHS of (8) with (A12) and simplifying the result obtained yield (10). $\square$

**Proof of Lemma 1.** We follow the approach presented in Lemma 3.1 of [16] to derive bounds for $c_k$ for $k = 1, 2, \ldots$ From Inequality (3.7) in Lemma 3.1, we set $\mu_0 = p = \frac{\nu}{2}$ and this immediately yields (12). The last statement of the lemma is true from Remark 3.1 of [16]. $\square$

**Proof of Theorem 3.** Using (11)–(14), we immediately obtain (15). Next, we define

$$P_K(\nu, \zeta) := \sum_{k=K+1}^{\infty} b_k(\gamma, \nu, \zeta) \tag{A13}$$

for $K \geq 0$ and $\nu > 0$, where $b_k(\gamma, \nu, \zeta)$ is given in (14).
   Applying Lemma 1, one can show that

$$\lim_{k \to \infty} \left| \frac{b_{k+1}(\gamma, \nu, \zeta)}{b_k(\gamma, \nu, \zeta)} \right| = \zeta < 1 \tag{A14}$$

providing $\beta > \frac{1}{2} \max_i \alpha_i$.
   Using the ratio test along with (A14), the infinite series $P_K(n, \zeta)$ converges absolutely for all $0 < \zeta < 1$ and $\nu > 0$. As a result, one can show from (13) that

$$\lim_{K \to \infty} B_{K,\infty}^{(\gamma)}(\zeta) = (2\beta)^\gamma e^{\frac{\delta}{2\zeta}} \lim_{K \to \infty} P_K(\nu, \zeta) = 0. \tag{A15}$$

   By utilizing (15) and (A15), we thus obtain (16). $\square$

**Proof of Theorem 4.** From Euler's transformation [31], we apply

$$_2\mathbf{F}_1(a, b; c; z) = (1-z)^{c-a-b}{}_2\mathbf{F}_1(c-a, c-b; c; z) \tag{A16}$$

for $c > a + b$ to the generalized hypergeometric functions on the RHS of (8) with $\gamma = m$ as follows:

$$_2\mathbf{F}_1\left(-k, 1-k-\frac{\nu}{2}; 1-k-\frac{\nu}{2}-m; z\right) = (1-z)^{k-m}{}_2\mathbf{F}_1\left(1-\frac{\nu}{2}-m, -m; 1-k-\frac{\nu}{2}-m; z\right). \tag{A17}$$

   Using (A17), it is easy to show that, when $k > m$,

$$_2\mathbf{F}_1\left(-k, 1-k-\frac{\nu}{2}; 1-k-\frac{\nu}{2}-m; 1\right) = 0 \tag{A18}$$

for all $m = 1, 2, \ldots$, and this completes the proof. $\square$

**Proof of Corollary 3.** We derived in the proof of Corollary 2 that the $c_k$'s satisfy (A12). By replacing the coefficients in the finite series on the RHS of (17) with (A12) and simplifying the result obtained, we immediately obtain (18). $\square$

**Proof of Theorem 5.** Set $X_i = \frac{Z_i^2}{\sigma_{(1,i)}^2}$. Thus, $X_i \sim \chi_1^2\left(\frac{\mu_{(1,i)}^2}{\sigma_{(1,i)}^2}\right)$ for all $i = 1, \ldots, n$. Moreover, $Y_{(1,n)}$ can be expressed as $Y_{(1,n)} = \sum_{i=1}^n a_{(1,i)}\sigma_{(1,i)}^2 X_i$. Utilizing Theorem 1, Theorem 2, and Theorem 4 with $\alpha_i = a_{(1,i)}\sigma_{(1,i)}^2$, $\nu_i = 1$, and $\delta_i = \left(\frac{\mu_{(1,i)}}{\sigma_{(1,i)}}\right)^2$ for all $i = 1, \ldots, n$, we immediately obtain that the PDF of $Y_{(1,n)}$, $\mathbb{E}[Y_{(1,n)}^\gamma]$, and $\mathbb{E}[Y_{(1,n)}^m]$ can be computed by using (3), (8), and (17), respectively, for all $\gamma \in \mathbb{R}_+$ and integer $m \in \mathbb{N}$. $\square$

**Proof of Theorem 6.** Let $\nu_i = 2\kappa_{(2,i)}$ and $X_i \sim \chi_{\nu_i}^2$. By using the property of the Gamma distribution, we have that $\frac{1}{2}\theta_{(2,i)}X_i \sim \mathbf{Gamma}\left(\kappa_{(2,i)}, \theta_{(2,i)}\right)$ and $G_i$ can be expressed as $G_i = \frac{1}{2}\theta_{(2,i)}X_i$, for all $i = 1, \ldots, n$. As a result, $Y_{(2,n)}$ can be expressed in terms of a linear combination of independent chi-square random variables as $Y_{(2,n)} = \sum_{i=1}^n \frac{1}{2}a_{(2,i)}\theta_{(2,i)}X_i$. Applying Theorems 1, 2, and 4 with $\alpha_i = \frac{1}{2}a_{(2,i)}\theta_{(2,i)}$, $\nu_i = 2\kappa_{(2,i)}$, and $\delta_i = 0$ for all $i = 1, \ldots, n$, the PDF of $Y_{(2,n)}$, $\mathbb{E}[Y_{(2,n)}^\gamma]$, and $\mathbb{E}[Y_{(2,n)}^m]$ can be computed by using (3), (8), and (17), respectively, for all $\gamma \in \mathbb{R}_+$ and integer $m \in \mathbb{N}$. $\square$

**Proof of Theorem 7.** Utilizing the property of the Erlang distribution, we have $L_i \sim \mathbf{Gamma}\left(\kappa_{(3,i)}, \theta_{(3,i)}\right)$, where $\theta_{(3,i)} = \frac{1}{\lambda_{(3,i)}}$ for all $i = 1, \ldots, n$. Applying Theorem 6, the PDF of $Y_{(3,n)}$, $\mathbb{E}[Y_{(3,n)}^\gamma]$, and $\mathbb{E}[Y_{(3,n)}^m]$ can be computed by using (3), (8), and (17), respectively, for all $\gamma \in \mathbb{R}_+$ and integer $m \in \mathbb{N}$, where we set $\alpha_i = \frac{a_{(3,i)}}{2\lambda_{(3,i)}}$, $\nu_i = 2\kappa_{(3,i)}$, and $\delta_i = 0$ for all $i = 1, \ldots, n$. $\square$

**Proof of Theorem 8.** Using the property of the exponential distribution, we have $P_i \sim \mathbf{Gamma}\left(\kappa_{(4,i)}, \theta_{(4,i)}\right)$, where $\kappa_{(4,i)} = 1$ and $\theta_{(4,i)} = \frac{1}{\lambda_{(4,i)}}$ for all $i = 1, \ldots, n$. From Theorem 6, the PDF of $Y_{(4,n)}$, $\mathbb{E}[Y_{(4,n)}^\gamma]$, and $\mathbb{E}[Y_{(4,n)}^m]$ can be computed by using (3), (8), and (17), respectively, for all $\gamma \in \mathbb{R}_+$ and integer $m \in \mathbb{N}$, where we set $\alpha_i = \frac{a_{(4,i)}}{2\lambda_{(4,i)}}$, $\nu_i = 2$, and $\delta_i = 0$ for all $i = 1, \ldots, n$. $\square$

**Proof of Theorem 9.** From the property of the Maxwell–Boltzmann distribution, we have $W_i^2 \sim \mathbf{Gamma}\left(\kappa_{(5,i)}, \theta_{(5,i)}\right)$, where $\kappa_{(5,i)} = \frac{3}{2}$ and $\theta_{(5,i)} = 2\phi_{(5,i)}^2$ for all $i = 1, \ldots, n$. Applying Theorem 6, the PDF of $Y_{(5,n)}$, $\mathbb{E}[Y_{(5,n)}^\gamma]$, and $\mathbb{E}[Y_{(5,n)}^m]$ can be computed by using (3), (8), and (17), respectively, for all $\gamma \in \mathbb{R}_+$ and integer $m \in \mathbb{N}$, where we set $\alpha_i = a_{(5,i)}\phi_{(5,i)}^2$, $\nu_i = 3$, and $\delta_i = 0$ for all $i = 1, \ldots, n$. $\square$

**Appendix C. Omitted Proofs from Section 4**

**Proof of Theorem 11.** First, we define

$$\hat{Y}_n := \sum_{i=1}^n \hat{\alpha}_i \hat{X}_i \tag{A19}$$

where the $\hat{X}_i$'s are the independent noncentral chi-square and chi-square random variables given in (23) with the coefficients and parameters given in (24)–(28).

Next, we use the Laguerre expansion (3) in Theorem 1 to obtain the PDF of $\hat{Y}_n$ by setting $\alpha_i = \hat{\alpha}_i$, $\nu_i = \hat{\nu}_i$, and $\delta_i = \hat{\delta}_i$ for $i = 1, \ldots, n$. As a result, the PDF of $\hat{Y}_n$, denoted by $f_{\hat{Y}_n}^{(\beta)}(\hat{y}_n)$, can be expressed as

$$f_{\hat{Y}_n}^{(\beta)}(\hat{y}_n) = \frac{e^{-\frac{\hat{y}_n}{2\beta}}\hat{y}_n^{\frac{\hat{v}_n}{2}-1}}{(2\beta)^{\frac{\hat{v}_n}{2}}} \sum_{k=0}^\infty \frac{k!}{\Gamma(\frac{\hat{v}_n}{2}+k)}\hat{c}_{k,n}\mathbf{L}_k^{\left(\frac{\hat{v}_n}{2}-1\right)}\left(\frac{\hat{y}_n}{2\beta}\right) \tag{A20}$$

for $\hat{y}_n > 0$ and $\beta > 0$, where

$$\hat{v}_n = \sum_{i=1}^{n} \hat{v}_i = \mathbf{d}(0) + \sum_{i=2}^{n} \mathbf{d}^{(1)}\left((i-1)\frac{t}{n}\right)\frac{t}{n}, \tag{A21}$$

$$\hat{c}_{0,n} = 1, \tag{A22}$$

$$\hat{c}_{k,n} = \frac{1}{k}\sum_{j=0}^{k-1} \hat{c}_{j,n}\hat{d}_{k-j,n}, \tag{A23}$$

for $k \geq 1$,

$$\begin{aligned}
\hat{d}_{1,n} &= -\frac{1}{2\beta}\hat{\delta}_1\hat{\alpha}_1 + \frac{1}{2}\hat{v}_1\left(1 - \frac{\hat{\alpha}_1}{\beta}\right) + \frac{1}{2}\sum_{i=2}^{n}\hat{v}_i\left(1 - \frac{\hat{\alpha}_i}{\beta}\right) \\
&= -\frac{1}{2\beta}v_0 e^{-\int_0^t \kappa(u)du} + \frac{1}{2}\mathbf{d}(0)\left(1 - \frac{\tau(t,0)}{\beta}\right) + \\
&\quad \frac{1}{2}\sum_{i=2}^{n}\mathbf{d}^{(1)}\left((i-1)\frac{t}{n}\right)\frac{t}{n}\left(1 - \frac{\tau\left(t,(i-1)\frac{t}{n}\right)}{\beta}\right),
\end{aligned} \tag{A24}$$

and

$$\begin{aligned}
\hat{d}_{j,n} &= -\frac{j}{2}\left(\frac{1}{\beta}\right)^j \hat{\delta}_1\hat{\alpha}_1(\beta - \hat{\alpha}_1)^{j-1} + \frac{1}{2}\hat{v}_1\left(1 - \frac{\hat{\alpha}_1}{\beta}\right)^j + \frac{1}{2}\sum_{i=2}^{n}\hat{v}_i\left(1 - \frac{\hat{\alpha}_i}{\beta}\right)^j \\
&= -\frac{j}{2}\left(\frac{1}{\beta}\right)^j v_0 e^{-\int_0^t \kappa(u)du}(\beta - \tau(t,0))^{j-1} + \frac{1}{2}\mathbf{d}(0)\left(1 - \frac{\tau(t,0)}{\beta}\right)^j \\
&\quad + \frac{1}{2}\sum_{i=2}^{n}\mathbf{d}^{(1)}\left((i-1)\frac{t}{n}\right)\frac{t}{n}\left(1 - \frac{\tau\left(t,(i-1)\frac{t}{n}\right)}{\beta}\right)^j,
\end{aligned} \tag{A25}$$

for $j \geq 2$.

It should be noted from (32)–(35) and (A22)–(A25) that choosing $\beta = \tau(t,0) > 0$ yields

$$\lim_{n\to\infty}\hat{v}_n = \mathbf{d}(t), \tag{A26}$$

$$\lim_{n\to\infty}\hat{d}_{j,n} = \hat{d}_j(t,v_0) \tag{A27}$$

for $j \geq 1$, and

$$\lim_{n\to\infty}\hat{c}_{k,n} = \hat{c}_k(t,v_0) \tag{A28}$$

for $k \geq 0$.

From Theorem 10, we apply the convergence (22) to $\hat{Y}_n$ as defined in (A19) and use (A26)–(A28) to obtain

$$v = \lim_{n\to\infty}\hat{y}_n, \tag{A29}$$

and

$$f_{V_t}(v,t|v_0) = \lim_{n\to\infty}f_{\hat{Y}_n}^{(\tau(t,0))}(\hat{y}_n) = \frac{e^{-\frac{v}{2\tau(t,0)}}v^{\frac{\mathbf{d}(t)}{2}-1}}{(2\tau(t,0))^{\frac{\mathbf{d}(t)}{2}}}\sum_{k=0}^{\infty}\frac{k!}{\Gamma(\frac{\mathbf{d}(t)}{2}+k)}\hat{c}_k(t,v_0)\mathbf{L}_k^{\left(\frac{\mathbf{d}(t)}{2}-1\right)}\left(\frac{v}{2\tau(t,0)}\right), \tag{A30}$$

respectively.

Furthermore, if $\mathbf{d}(s) = d \geq 2$ for all $s \in [0,t]$ then $\mathbf{d}^{(1)}(s) = 0$ for all $s \in [0,t]$. Hence, $\hat{c}_k(t,v_0)$ for $k \geq 0$ as written in (32) and (33) are simplified to (36) by replacing $\mathbf{d}^{(1)}(s)$ in (34) and (35) with zero, and the proof is now complete. $\square$

**Proof of Theorem 12.** According to the proof of Theorem 11, we first apply Theorem 2 to $\hat{Y}_n$ defined in (A19) to obtain

$$
\begin{aligned}
\mathbb{E}^{\mathbb{P}}\big[\hat{Y}_n^{\gamma}\big] &= \int_0^{\infty} \hat{y}^{\gamma} f_{\hat{Y}_n}^{(\beta)}(\hat{y}) d\hat{y} \\
&= (2\beta)^{\gamma} \sum_{k=0}^{\infty} (-1)^k \frac{\Gamma\left(\gamma + k + \frac{\hat{v}_n}{2}\right)}{\Gamma\left(k + \frac{\hat{v}_n}{2}\right)} {}_2\mathbf{F}_1\left(-k, 1 - k - \frac{\hat{v}_n}{2}; 1 - k - \frac{\hat{v}_n}{2} - \gamma; 1\right) \hat{c}_{k,n}.
\end{aligned}
\tag{A31}
$$

By choosing $\beta = \tau(t, 0) > 0$ and using the results written in (A26)–(A30), an explicit formula for $U_E^{(\gamma)}(t|v_0) = \lim_{n \to \infty} \mathbb{E}^{\mathbb{P}}\big[\hat{Y}_n^{\gamma}\big]$ is obtained as expressed in (38). In addition, when $m \in \mathbb{N}$, an explicit formula for $U_E^{(m)}(t|v_0)$ can be obtained as expressed in (39) by applying Theorem 4 to (38), and this completes the proof. $\square$

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
