# Peer review of "Analytically Computing the Moments of a Conic Combination of Independent Noncentral Chi-Square Random Variables and Its Application for the Extended Cox–Ingersoll–Ross Process with Time-Varying Dimension"

_mathematics, doi:10.3390/math11051276_

Round 1

Reviewer 1 Report

Authors had transformed asymptotic expansions for the distributions of the weighted sums of independent chi-square random variables obtained by other authors in reference [20] into analogous expansions for order $\gamma>0$ moments of such sums. Estimates of accuracy of moments by truncated series are obtained in analytical and numerical forms. 

I have read the article "along the diagonal", it seems for me appropriate for publication. 

It should be noted that the sentence before Theorem 2 on the page 3 is incomplete. 

Reviewer 2 Report

The paper is well-written, the results applicable and hence, in my opinion, suitable for publication. However, to enhance the discussion, I suggest extending 4 with more details about papers where explicit formulas for conditional expectation were derived (e.g. [46]), and, if possible comparing with novel results.

Reviewer 3 Report

Using the Laguerre expansion of PDFs, the authors are able to express the moments of a class of random variables via a convergent series based on generalized hypergeometric functions. Such class of random variables consists of linear combination (with positive weights) of independent variables of the type Z^2G. A list of examples of Z and G have been studied. An interesting application is made, for evaluating the moments of extended CIR models.

The paper is mathematically correct and well written. The statements, the proofs, the simulation study are complete. The main contribution is the derivation of the rth moment of the random variable. This contribution indeed fills some gaps when evaluating the moments of ECIR.

I recommend accepting the paper subject to minor review:

(1) The title of the paper is too long. It is great f the authors could cut it short.

(2) Throughout the paper I suggest to set gamma>0 and m>0. The cases for gamma=0 and m=0 does not need being studied. Otherwise it will take more efforts to explain why the RHS of (8) is 1 when gamma=0.

(3) Line 104, it is better o recall the definition of the key concept the generalized hypergeometric function in the paper. Once the function is defined in the paper, it can be referred to obtain (A18).

(4) Line 107, ck are not arbitrary, in fact they should follow (5)-(7). The same corrections should be made for Corollary 1 and Theorem 4.

(5) Line 165, in in.

(6) Line 179, Considers

(7) Line 516, I am not aware of proving statement using MATEMATICA. May the authors prove (A2) using the properties of generalized hypergeometric functions directly?

(8) The second line of Equation (A24), split it into 2 lines.

Finally I would ask the authors 2 questions. Such discussion is not necessarily included in the paper:

(1) Is there a general form of PDFs such that their moments can be represented as (3), (8), (17)? The current paper only lists some examples of PDFs.

(2) By similar approach, is there a way to derive the cross-moments of ECIR, i.e., the covariance function of ECIR?

Reviewer 4 Report

Report

February 10th, 2023

Paper

Analytically computing the moments of a conic combination of independent noncentral chi-square random variables and its application for the extended Cox–Ingersoll–Ross process with time-varying dimension

Sanae Rujivan, Athinan Sutchada, Kittisak Chumpong, Napat Rujeerapaiboon

Mathematics-2216573 (paper number)

Mathematics

Summary of the paper from the reviewer

The topic of the paper is interesting especially because of its application in modelling of interest rates and pricing of financial products. Here is investigated the γ-th moment and γ-th conditional moment of random variable Yn defined as a sum of various types of random variables with noncentral chi-square distributions. In the paper are presented 4 examples with MC simulations. The results are obtained by Mathematica.

In the introduction is not clearly enough presented achievement of this research. However, the introduction needs to make it more clear what the main objective of the paper is, and what the contribution is compared to existing literature.

You introduce a random variable as a sum of infinity many random variables (set Z of integer number is denumerable) with chi-square distribution, which means that you are talking about absolutely continuous random variables, without introducing a probability space where the random variables are defined. Also you are not explaining is there a need that all of random variables should be defined on the same probability space or not.

All results are generally an application on some special cases of random variables and special cases of PDF from [20]  and [35]. In almost all theorems and lemmas you are giving an explicit formula for E[Yγ], without defining the conditions for existence? Why?

The paper cannot be published in its current form and I recommend MAJOR REVISION.

Round 2

Reviewer 4 Report

Report

March 3rd, 2023

Paper

Analytically computing the moments of a conic combination of independent noncentral chi-square random variables and its application for the extended Cox–Ingersoll–Ross process with time-varying dimension

Sanae Rujivan, Athinan Sutchada, Kittisak Chumpong, Napat Rujeerapaiboon

Mathematics-2216573 (paper number)

Mathematics

Summary of the paper from the reviewer

The topic of the paper is interesting especially because of its application in modelling of interest rates and pricing of financial products. Here is investigated the γ-th moment and γ-th conditional moment of random variable Yn defined as a sum of various types of random variables with noncentral chi-square distributions. In the paper are presented 4 examples with MC simulations. The results are obtained by Mathematica.

The paper is very much improved.

The paper can be published in its current form and I recommend ACCEPTANCE
